# A toolbox of engineered mosquito lines to study salivary gland biology and malaria transmission

**Dennis Klug** ⓘ *, **Katharina Arnold, Raquel Mela-Lopez, Eric Marois, Stéphanie A. Blandin**

Université de Strasbourg, CNRS UPR9022, INSERM U1257, Institut de Biologie Moléculaire et Cellulaire, Strasbourg, France

* dennis.klug@sciencebridge.net

**Data Availability Statement:** All relevant data are included in the manuscript and supporting information files. Transgenic mosquito lines as well as plasmids are available from resource centers

## Abstract

Mosquito saliva is a vehicle for the transmission of vector borne pathogens such as *Plasmodium* parasites and different arboviruses. Despite the key role of the salivary glands in the process of disease transmission, knowledge of host-pathogen interactions taking place within this organ is very limited. To improve the experimental tractability of the salivary glands, we have generated fluorescent reporter lines in the African malaria mosquito *Anopheles coluzzii* using the salivary gland-specific promoters of the *anopheline antiplatelet protein* (AAPP), the *triple functional domain protein* (TRIO) and *saglin* (SAG) coding genes. Promoter activity was specifically observed in the distal-lateral lobes or in the median lobe of the salivary glands. Besides a comparison of the expression patterns of the selected promoters, the fluorescent probes allowed us to evaluate the inducibility of the selected promoters upon blood feeding and to measure intracellular redox changes. We also combined the *aapp-DsRed* fluorescent reporter line with a pigmentation-deficient *yellow(-)* mosquito mutant to assess the feasibility of *in vivo* microscopy of parasitized salivary glands. This combination allowed locating the salivary gland through the cuticle and imaging of individual sporozoites *in vivo*, which facilitates live imaging studies of salivary gland colonization by *Plasmodium* sporozoites.

## Author summary

Over the course of evolution, a number of pathogens have developed the ability to replicate within and to be transmitted by mosquitoes. An important adaptation of pathogens to be successfully transmitted is their ability to colonize mosquito slivary glands, as the timing and efficiency with which pathogens enter the glands directly affects their transmission. Despite the importance of salivary glands in disease transmission, host-pathogen interactions in this organ are poorly understood. To facilitate biological studies, we developed fluorescent reporter lines in the malaria mosquito *Anopheles coluzzii* by placing fluorescent probes under the control of the salivary gland-specific promoters of the *anopheline antiplatelet protein* (AAPP), *triple functional domain protein* (TRIO), and *saglin* (SAG) genes. We used these lines to characterize inducibility by blood meals and

(MR4 and Addgene) or upon request from IBiSA (Infrastructures en Biologie Santé et Agronomie) https://www.ibisa.net/plateformes/insectarium-631.html#mapid, lramolu@unistra.fr.

**Funding:** This work was supported by a grant from Laboratoire d'Excellence (LabEx) ParaFrap (ANR-11-LABX-0024) and an ERC Starting Grant (N° 260918) to SB as well as by a grant from Equipement d'Excellence (EquipEx) I2MC (ANR-11-EQPX-0022) to EM. DK was funded by a DFG postdoctoral fellowship (KL 3251/1-1) and RML received a PhD fellowship from the University of Strasbourg IdEx program (Unistra IdEx fellowship 2015). The funders had no role in study design, data collection and analysis, decision to publish, or preparation of the manuscript.

**Competing interests:** The authors declare that they have no conflict of interest.

spatial expression of the different promoters, as well as intracellular redox changes in salivary glands after sporozoite invasion. In addition, we combined the *aapp-DsRed* fluorescent reporter line with a pigmentation-deficient *yellow(-)* mosquito mutant to evaluate the feasibility of *in vivo* microscopy of parasitized salivary glands. Our data suggest that salivary gland invasion is an active process that requires *Plasmodium* sporozoites to actively move through the tissues of the mosquito to reach the salivary glands.

## Introduction

Vector borne diseases have been a major health burden in the history of mankind. An efficient way to counteract diseases transmitted by blood sucking arthropods is to prevent transmission by reducing the number of vectors in the environment or by preventing vectors from biting humans in risk areas. As a consequence of malaria control interventions, approximately 663 million clinical malaria cases could be averted between 2000 and 2015. The most effective measure was the use of insecticide treated bet nets, which prevented an estimated 68% of cases [1]. Still, preventive measures are counteracted by the very effective transmission of vector borne pathogens, which hampers eradication approaches. The key organ for transmission of vector-borne diseases is the salivary gland, as pathogens must colonize this organ to subsequently enter the saliva and be deposited in the skin during a blood meal. Host-pathogen interactions that take place in this organ are crucial for the spread of pathogens, as the concentration of pathogens in the mosquito saliva, as well as the composition and quantity of the saliva itself, are critical for successful transmission. To successfully colonize salivary glands, different pathogens have developed different strategies. While vector-borne viruses such as DENV cause systemic infection of the mosquito and replicate in the salivary glands [2], malaria parasites of the genus *Plasmodium* proliferate exclusively in the mosquito midgut. After infection of the mosquito midgut through the uptake of a blood meal from an infected host, *Plasmodium* parasites form cysts, called oocysts, below the basal lamina of the midgut epithelium where they undergo multiple rounds of DNA replication. Rapid cell division, called sporogony, then follows, leading to the generation of several thousands of sporozoites per oocyst. These cells are released from oocysts and travel from the midgut to the salivary glands that they specifically recognize and invade. After being deposited along with mosquito saliva in the skin during a blood meal, sporozoites traverse the dermis, enter blood capillaries and reach the human liver where they pursue development. On the path of sporozoites from the mosquito midgut to the liver of its intermediate host, the salivary glands represent a special niche in which they mature before being transferred to the host with the mosquito saliva.

 Although the general life cycle of *Plasmodium* is well known, we lack detailed knowledge of the temporal relationships and host-parasite interactions that occur in the various tissues and cells which the parasite, particularly the sporozoite, traverses and invades. While it is believed that the invasion process of the salivary glands by sporozoites that have egressed from oocysts is rapid (~1–10 minutes), *in vivo* data supporting this assumption is lacking [3]. Still, the time a sporozoite needs to reach and successfully invade the glands is an important parameter, because at this timepoint, a mosquito is considered infective to humans. In addition, microscopic examinations of infected glands revealed that sporozoites form large aggregations in organized bundles in the cavities of the acinar cells [4]. The degree of order in these structures raises the question of whether the sporozoites recognize each other and possibly enter a quiescent phase to bridge the time until the vector next blood meal, or whether these bundles form by mechanical constraints alone. Based on the accumulation of sporozoites in specific sub-regions of

salivary glands, it has also been hypothesized that sporozoites may preferentially invade the gland at specific sites [5]. However, since no receptor-ligand interaction between sporozoites and acinar cells has yet been clearly identified, the detection of such entry sites is difficult to prove. The study of gland-sporozoite or sporozoite-sporozoite interactions in the gland has generally been limited to the observation of infected tissues *ex vivo* because the highly pigmented cuticle of the mosquito makes microscopy of parasite stages in living mosquitoes difficult and, at the same time, an *in vitro* culture system for the insect stages of *Plasmodium* is not available. While *ex vivo* studies significantly contributed to the knowledge about *Plasmodium* we have today, samples obtained in this way are less suitable for imaging dynamic processes because dissected tissues change and decompose rapidly. In addition, the study of processes taking place in salivary glands has been complicated by the lack of molecular tools for mosquitoes like fluorescent reporter lines with tissue-specific gene expression, which have been slow to develop due to difficulties in transgenesis. The implementation of CRISPR/Cas9 technology in mosquito research has been a decisive step forward, which will give access to the analysis of the molecular mechanisms underlying mosquito-pathogen interactions at an unprecedented level of details. Females of *Aedes spp*. and *Anopheles spp*. mosquitoes possess two salivary glands each consisting of two lateral lobes and one median lobe. In addition, the lateral lobes are further subdivided into a distal-lateral and a proximal-lateral part which differ visually in cell size and in the size of formed cavities in which cells secrete saliva components [6]. Transcriptomic studies have shown that salivary glands display a tissue-specific expression profile in which many proteins thought to have a function in blood uptake are indeed absent in male salivary glands [7]. Consistent with the reduction in expressed proteins, male salivary glands are smaller and differ in their morphology, consisting of only one lobe per gland. Detailed knowledge about the expression of proteins in the salivary glands could be obtained by applying *ex vivo* approaches. *In situ* hybridization of gland-specific transcripts on dissected *Aedes aegypti* salivary glands has shown that genes are spatially expressed in the median, distal-lateral, and proximal-lateral lobes or combinations thereof [8], suggesting that different sections of the salivary gland may provide different components of mosquito saliva. In addition, *Plasmodium* sporozoites have been shown to invade primarily the distal-lateral lobes [5], which has led to speculation about a specific protein interface used by sporozoites to recognize the distal-lateral sections.

Recombinant salivary proteins considered to have therapeutic potential have been produced and studied *in vitro* for their vasodilator, anticoagulant and anti-inflammatory properties. Examples include aegyptin [9], anopheline [10], and anopheline anti-platelet protein (AAPP) [11], all of which act as anticoagulants. From this selection, AAPP is one of the few proteins that have also been studied *in vivo*. In *A. stephensi*, the generation of one of the first transgenic mosquito lines showed that the *aapp* promoter drives *DsRed* expression exclusively in the distal-lateral lobes [12]. In addition to their primary physiological function, salivary gland proteins suspected to play a role in the spread of pathogens such as the malaria parasite *Plasmodium* have been the focus of several studies. Protein-protein interaction-based approaches using sporozoite surface proteins and salivary gland proteins identified the proteins saglin and CSPBP as potential determinants for sporozoite invasion [13,14]. Knockdown experiments of both proteins have shown that sporozoite numbers in the salivary glands are reduced compared to controls, indicating a lower invasion rate. However, overexpression of saglin in the distal-lateral lobes of *A. stephensi* did not result in an increase in the number of sporozoites colonizing the salivary glands, raising doubt about the importance of saglin in this process [15]. Another protein that has been shown to affect sporozoite transmission is AgTRIO (triple functional domain protein), that, like saglin, belongs to the SG1-protein family [7,16]. Active and passive immunization of mice with recombinant AgTRIO was shown to diminish transmission of *P. berghei* sporozoites [17,18]. Interestingly, TRIO expression

increases in the presence of *P. berghei* sporozoites, suggesting that sporozoites in salivary glands may influence protein expression of their host cells to potentially increase transmission efficiency [17]. Here we generated and characterised three fluorescent salivary gland reporter lines using the upstream regions of the *aapp*, *trio* and *saglin* genes in the African malaria vector *Anopheles coluzzii*, and we used one of them in combination with a pigment-deficient line to perform *in vivo* live imaging of sporozoites invading salivary glands. We expect the lines described here will be of use to investigate salivary gland and sporozoite biology.

## Results

### Generation of transgenic mosquito lines with salivary gland-specific reporter expression and low pigmentation

To generate salivary gland reporter lines, the 5' upstream sequences of *saglin* (AGAP000610), *aapp* (AGAP009974) and *trio* (AGAP001374) were used to drive the expression of DsRed and hGrx1-roGFP2 from transgenic constructs integrated in the *X1* docking line (**Figs 1A and S1A** and **S1 Appendix**) with an *attP* site on chromosome 2L [19]. The fluorescence reporter *hGrx1-roGFP2* was chosen because it bears the advantages of EGFP and enables the measurement of the oxidation level of the intracellular glutathione pool by ratiometric quantification [20]. We used the *aapp* promoter sequence from *Anopheles coluzzii* orthologous to the one used to drive salivary gland-specific transgene expression in *Anopheles stephensi* [12]. For both *saglin* and *trio*, we defined putative promoter regions according to the genomic context. In addition, we also used CRISPR/Cas9 to integrate a fluorescence cassette encoding EGFP into the endogenous *saglin* locus, replacing the coding sequence, to study *saglin* promoter activity in its native genomic context. Indeed, activity of the cloned promoter selected for ectopic expression was uncertain due to the shortness of the apparent *saglin* promoter region (220bp, **S2 Fig**). For this, site directed knock-in mutagenesis was performed by injecting a plasmid encoding three guide RNAs targeting *saglin* and carrying a repair template marked with *3xP3-EGFP* into mosquito eggs from a line with germline-specific expression of Cas9 (*vasa-Cas9*), generating a *sag(-)KI* line (**Figs 1B and S3A**). To place the *EGFP* gene directly under control of the *saglin* promoter, the *3xP3* promoter used to select transgenic mosquito larvae was flanked by lox sites to enable its excision by Cre recombination (**S3A Fig**, Material & Methods). Similarly, we generated a knockin (KI) into the *yellow* gene that encodes a protein required for the synthesis of black pigments in insects [21,22]. For this, we used a plasmid with guide RNAs specific for *yellow* and corresponding homology arms flanking a fluorescence cassette expressing *EGFP* surrounded by lox sites (**S3B Fig**). Through crossing of salivary gland reporter lines with low pigmented *yellow(-)KI* mosquitoes, we aimed to improve the fluorescence signal of the expressed reporter and extend the range of applications, e.g. for *in vivo* imaging. Genotyping by PCR confirmed the successful generation of the salivary gland reporter lines *aapp-DsRed*, *aapp-hGrx1-roGFP2*, *trio-DsRed* and *sag-DsRed* (**S1B Fig**) as well as the mosquito lines *sag(-)KI* and *yellow(-)KI* (**S3C and S3D Fig**). Loss of the *3xP3* promoter in *sag(-)EX* mosquitoes (mosquitoes lacking the *3xP3* promoter after *Cre* mediated excision) was confirmed by PCR in the progeny of a cross between the *sag(-)KI* line and a mosquito line expressing *Cre* recombinase in the germline (**S3E Fig**) [19].

### Lobe-specific transcriptional activation by *sag*, *aapp* and *trio* promoter sequences

Dissection of female salivary glands revealed a highly specific expression pattern of DsRed, EGFP and hGrx1-roGFP2 driven by *sag*, *aapp* and *trio* promoter sequences inside salivary

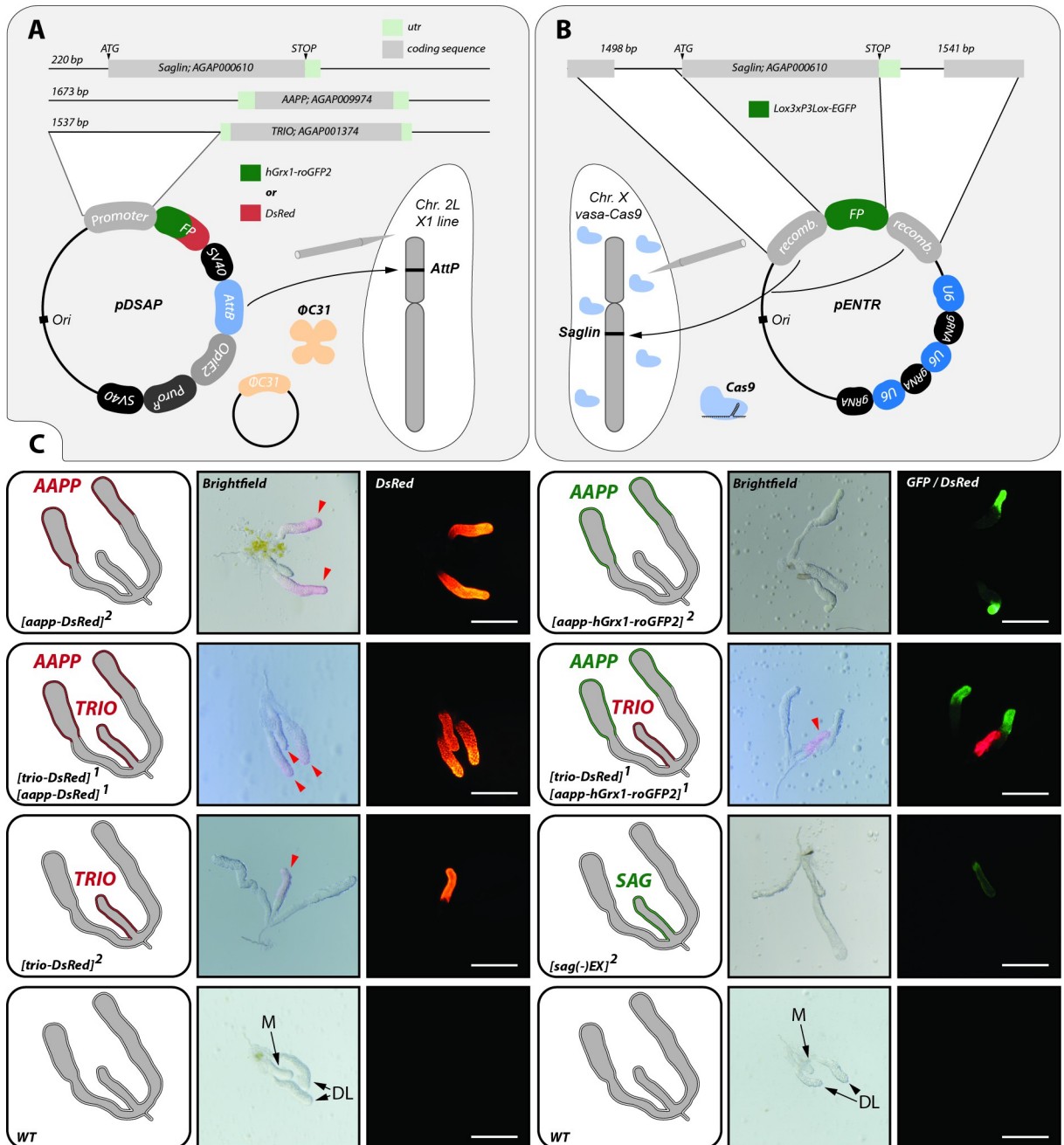

**Fig 1. Lobe-specific reporter expression observed in female *aapp-DsRed*, *aapp-hGrx1-roGFP2*, *trio-DsRed* and *sag(-)EX* mosquitoes. A)** ΦC31 integrase-mediated transgene insertion. The *aapp*, *trio* and *saglin* promoter sequences were fused to *DsRed* or *hGrx1-roGFP2* within the *pDSAP* vector which enables chemical selection of mosquito larvae by puromycin. Transgenes were integrated by ΦC31 integrase-mediated recombination into the *X1* line harbouring an *attP* site on chromosome 2L. **B)** Cas9-mediated knockin of a fluorescence cassette in the *saglin* gene (AGAP000610). Upstream and downstream sequences of *saglin* were cloned into the *pENTR* vector flanking a fluorescence cassette expressing *EGFP* under the control of the *3xP3* promoter. Three guides targeting *saglin* controlled by *U6* promoters were encoded on the same *pENTR* vector. Transgenesis was performed in mosquitoes with germline-specific expression of *Cas9* (*vasa-Cas9*). **C)** Fluorescent reporter expression in mosquitoes homozygous for *trio-DsRed*, *aapp-DsRed*, *aapp-hGrx1-roGFP2* and *sag(-)EX* (*3xP3* promoter excised) as well as in females heterozygous for *trio-DsRed* and *aapp-DsRed*, and *trio-DsRed* and *aapp-hGrx1-roGFP2*. Salivary glands of wild type (*G3*) are shown as controls. Illustrations on the left of each panel indicate the genotype of the gland as well as the observed fluorescence pattern. Images of salivary glands are shown in brightfield beside an image of the same gland in the relevant fluorescence channel. Red arrows in brightfield images highlight the pink parts of *aapp-DsRed* and *trio-DsRed* salivary glands in the absence of fluorescence excitation indicating the high expression of DsRed. Median (M) and distal-lateral lobes (DL) are indicated on the images of wild type glands. Salivary glands were dissected from 2–10 day old females that had not previously received blood feeding. Scale bars: 250 μm.

glands (**Fig 1C**). While no fluorescence was observed for DsRed driven by the cloned upstream sequence of the *saglin* gene in dissected salivary glands (**S2A Fig**), promoter activity was detected in the line where *EGFP* replaced the *saglin* gene (**Fig 1C**), indicating that essential transcriptional elements localize outside of the intergenic sequence between *AGAP000609* and *AGAP000610* (*saglin*) (**S2B Fig**). Interestingly, fluorescence signals in glands of female *aapp-DsRed*, *aapp-hGrx1-roGFP2* mosquitoes were restricted to the distal-lateral lobes, while female *trio-DsRed* and *sag(-)EX* mosquitoes displayed fluorescence exclusively in the median lobe (**Fig 1C**). This difference became especially evident when crossing *aapp-hGrx1-roGFP2* mosquitoes with *trio-DsRed* mosquitoes giving rise to a trans-heterozygous offspring displaying hGrx1-roGFP2 expression in distal-lateral lobes combined with DsRed expression in the median lobe (**Fig 1C**). Similarly, when crossing *aapp* and *trio* DsRed reporter lines, the entire salivary gland was red fluorescent in the F1 progeny (**Fig 1C**). Expression of *trio-DsRed* and *aapp-DsRed* was so strong that the respective parts of the salivary gland appeared pink colored even in brightfield microscopy (**Fig 1C**). Of note, fluorescent reporter expression in the salivary glands of *aapp-DsRed*, *aapp-hGrx1-roGFP2* and *trio-DsRed* mosquito colonies was restricted to females while no fluorescence signal could be observed in this organ in males (**S4A Fig**). Imaging of *aapp-DsRed*, *aapp-hGrx1-roGFP2* and *trio-DsRed* pupae during development revealed no distinct fluorescence signal in the head part where the salivary glands are located (**S4B Fig**). Still, *DsRed* expression was visible in the salivary glands of *aapp-DsRed* females that had been dissected directly after emergence (**Fig 2A**), indicating that the *aapp* promoter is already active in pharate adults. In contrast, no *DsRed* expression was observed in the glands of freshly hatched *trio-DsRed* females (**Fig 2A**). To evaluate the expression profile over time we performed a time-course experiment by dissecting and imaging salivary glands of *trio-DsRed* and *aapp-DsRed* females every day. *DsRed* expression controlled by *aapp* and *trio* promoters increases steadily until day 5 after hatching (**Fig 2B**). The increase in *DsRed* expression over time in the salivary glands of *aapp-DsRed* and *trio-DsRed* females could also be followed in transmitted light based on the pink coloration of the distal-lateral lobes (**S4C Fig**). In *Anopheles stephensi*, *aapp* expression was shown by qRT-PCR to be upregulated upon blood feeding [12]. We followed DsRed fluorescence levels in the salivary glands of *aapp-DsRed* and *trio-DsRed* sibling mosquitoes that were fed on blood or kept on sugar (**Fig 2C and 2D**). Our results confirm that the *aapp* promoter is induced upon ingestion of a blood meal while there was no difference between blood-fed and sugar-fed in females expressing the *trio-DsRed* transgene.

## Spatial expression pattern of the *saglin* and *trio* promoters in the median lobe

While expression of the fluorescent probes *DsRed* and *hGrx1-roGFP2* in the mosquito lines *aapp-DsRed*, *trio-DsRed* and *aapp-DsRed* is driven by the cloned *aapp* and *trio* promoter sequences, the selection of transgenic *sag(-)KI* larvae required the use of the synthetic *3xP3* promoter to allow screening for expression of *EGFP* in neonate larvae. To assess the activity of the native *saglin* promoter, the *3xP3* promoter was flanked by lox sites that allowed its excision using Cre to generate the *sag(-)EX* line (**S3E Fig**). Imaging of salivary glands (dissected tissues or through the cuticle in live mosquitoes) from *sag(-)KI* and *sag(-)EX* females revealed that *EGFP* expression is much stronger in specimens expressing the *sag(-)KI* transgene, indicating that the synthetic *3xP3* promoter enhances the native transcriptional activity of the *saglin* promoter (**Fig 3A**). As for the *aapp* promoter, reporter expression in *Sag(-)EX* was detected in salivary glands of mosquitoes that had just emerged (**Fig 2A**), and increased after blood feeding (**Fig 2C and 2D**). Of note, EGFP signal was 1.8 and 2.6 fold stronger at the base than at the

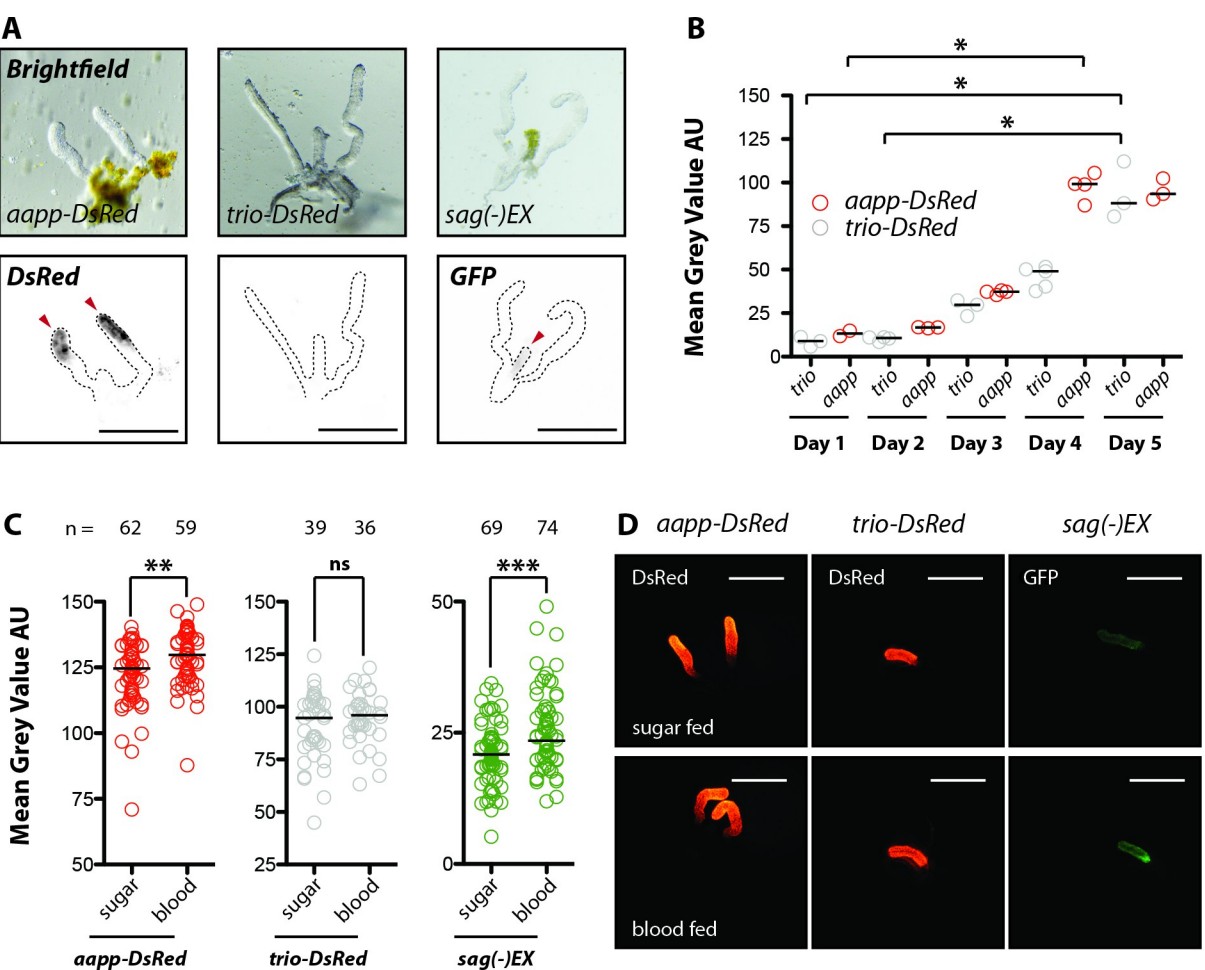

**Fig 2. Onset, strength and inducibility by blood feeding of reporter expression in *aapp-DsRed*, *trio-DsRed* and *sag(-)EX* mosquitoes. A)** Images of salivary glands dissected directly after mosquitoes emerged. Fluorescence signals were observed in salivary glands expressing *aapp-DsRed* and *sag(-)EX* (marked by red arrowheads) but not *trio-DsRed*. Scale bar: 250 μm. **B)** Mean fluorescence intensity measurements of dissected *aapp-DsRed* and *trio-DsRed* salivary glands from day one to day five after hatching. Data were normalized for the highest exposure time. Each dot represents a single gland (2–5 glands / timepoint) and bars the median. *p<0.01, one-way-ANOVA (Dunn's Multiple Comparison Test). **C)** The blood meal inducibility of the *trio*, *aapp* and *saglin* promoters was assessed by providing sibling females either with a blood meal or sugar solution. Approximately 24h later, salivary glands were dissected, imaged and the mean DsRed or EGFP fluorescence intensities were quantified. Data pooled from three (*aapp-DsRed* and *trio-DsRed*) or four (*sag(-)EX*) independent experiments. Horizontal black bars represent the median. **p<0.001 and *** p<0.0001, Mann Whitney test. ns: not significant. **D)** Images of salivary glands dissected 24h after blood feeding in comparison to sugar fed controls. Scale bar: 250 μm.

apex of the median lobe of the salivary gland in *sag(-)KI* and *sag(-)EX* females, respectively (**Fig 3A and 3B**). The mean fluorescence of *sag(-)KI* was 5.5-fold and 3.8-fold higher than that of *sag(-)EX* at the apex and base, respectively (**Fig 3B**). Next, we crossed *sag(-)KI* and *trio-DsRed* mosquitoes. Confocal imaging of salivary glands dissected from heterozygous individuals expressing both transgenes revealed a striking difference in the expression pattern of the *trio* and *saglin* promoters. *DsRed* expression driven by *trio* was observed strongest in the apex of the median lobe with low to no expression in cells of the base (**Fig 3C and 3D**). In contrast *EGFP* expression driven by the *saglin* promoter was detectable along the median lobe in an opposite gradient (**Fig 3B and 3D**). Imaging of the base of a gland expressing *trio-DsRed* and *sag(-)KI* showed a remarkably sharp expression boundary between cells with high and low *saglin* promoter activity. Only few high producer cells close to the boundary were observed to

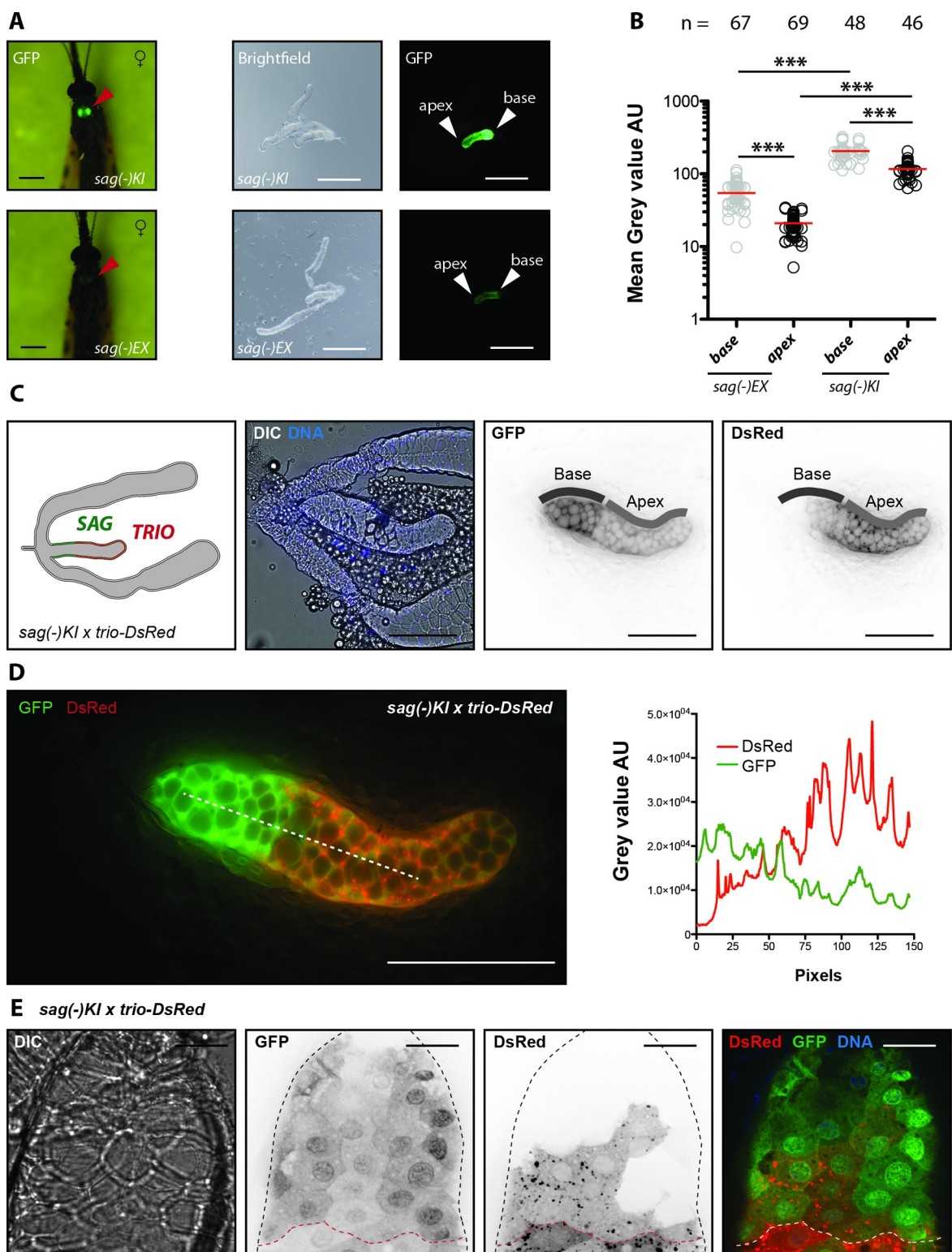

**Fig 3. Fluorescent reporter expression in the median lobe reveals spatial expression differences of *saglin* and *trio* promoters. A)** *EGFP* expression in homozygous *sag(-)KI* and *sag(-)EX* females seven days after emergence (left side). Red arrows point towards the location of the salivary glands. Scale bar: 0.5 mm. Brightfield and GFP signal in dissected salivary glands of homozygous *sag(-)KI* and *sag (-)EX* females (right side). Positions of the base and apex of the median lobe are indicated. Scale bar: 250 μm. **B)** Mean EGFP fluorescence quantification at the apex and at the base of the median lobe of *sag(-)EX* and *sag(-)KI* salivary glands dissected after sugar feeding. The

number of measured salivary glands is indicated above each column. Data from four (*sag(-)KI*) and three (*sag(-)EX*) experiments obtained from two different mosquito generations. The black line indicates the median. ***p<0.0001, one-way-ANOVA (Bonferroni's Multiple Comparison Test). **C)** Optical (C, D) and confocal (E) imaging of salivary gland heterozygous for *sag(-)KI* and *trio-DsRed*. Left to right (C): Illustration of the genotype and observed fluorescence pattern, differential interference contrast (DIC) and nuclear staining with Hoechst 33342, GFP signal and DsRed signal as black on white. Positions of the base and apex of the median lobe are indicated. Scale bar: 100 μm. **D)** Merge of the GFP and DsRed signals shown in (C). Line plot on the right shows measured GFP and DsRed signals along the dotted white line. Scale bar: 100 μm. **E)** Confocal imaging of a salivary gland heterozygous for *sag(-)KI* and *trio-DsRed*. The basal area of the median lobe is shown. The black dotted line indicates the outline of the salivary gland as seen in DIC while the red dotted line indicates the border between cells displaying high and low *saglin* promoter activity. Scale bar: 20 μm.

co-express *trio-DsRed* (**Fig 3E**). Using z-stacks of confocal images from dissected *sag(-)KI* salivary glands stained with Hoechst 33342, we further determined that the median lobe consists of 144 cells on average, of which 49 cells display high and 95 low *saglin* promoter activity. Considering the observed differences in the mean fluorescence between apex and base and assuming that the expression of *EGFP* represents the expression of *saglin*, cells at the base of the median lobe contribute ~88% to Saglin expression, while the cells in the apex contribute only ~12%, although the apex area contains nearly twice as many cells as the base.

## Subcellular localization of fluorescent reporters

During first microscopic examinations using low magnification, different intracellular localizations of the fluorescent proteins DsRed, EGFP and hGrx1-roGFP2 were observed (**Fig 1C**), which was confirmed by confocal microscopy. DsRed expressed in the salivary glands of *aapp-DsRed* and *trio-DsRed* mosquitoes appeared spotty and displayed a vesicle-like pattern within the cytoplasm of acinar cells, while hGrx1-roGFP2 and EGFP expressed in the glands of *aapp-hGrx1-roGFP2* and *sag(-)EX* mosquitoes displayed a more cytoplasmic localization (**Fig 4**). Due to the anatomy of the secretory cells, which form an inward cavity into which saliva is secreted, the vesicle-like DsRed accumulations localized mainly at the outer rim of the salivary gland. In contrast, localization of hGrx1-roGFP2 and EGFP was homogenous in the cytoplasm but appeared to be enriched around nuclei (**S5 Fig**).

## Variation of DsRed expression in *trio-DsRed* mosquitoes

During the generation of the salivary gland reporter lines and their subsequent outcrossing we observed that most L1 mosquito larvae carrying the *trio-DsRed* transgene displayed fluorescence in the eyes (**Fig 5A**), which was absent in the two other reporter lines *aapp-DsRed* and *aapp-hGrx1-roGFP2*. Interestingly, a few L4 stage larvae of the *trio-DsRed* line even displayed strong ubiquitous *DsRed* expression (**Fig 5A**). At the pupal stage, *DsRed* expression was detected in the ocellus, an eye spot close to the main compound eye, the antennae and/or throughout the body. The observed expression patterns were highly variable and ranged from completely non-fluorescent pupae to pupae expressing *DsRed* in all three tissues (**S6A Fig**). In adult mosquitoes, DsRed signal was less distinct compared to pupae although complete body fluorescence was occasionally observed in both males and females, with the strongest expression of *DsRed* in the salivary glands of females and the palps and antennae of males (**Fig 5B**). Dissections of adult mosquitoes displaying whole body fluorescence at the pupal stage revealed strong and sex-independent DsRed signals in Malphigian tubules, an insect organ that performs functions similar to the vertebrate kidney (**Fig 5C**). Of note, fluorescence in tubules strictly correlated to the high pupal fluorescence phenotype and was never observed in mosquitoes hatched from low fluorescent pupae (**Fig 5D**). We next investigated sex-specificity of each fluorescence pattern at the pupal stage and inheritance of the high and low fluorescent *trio-DsRed* transgene. Non-fluorescent pupae were always females, while fluorescent antennae

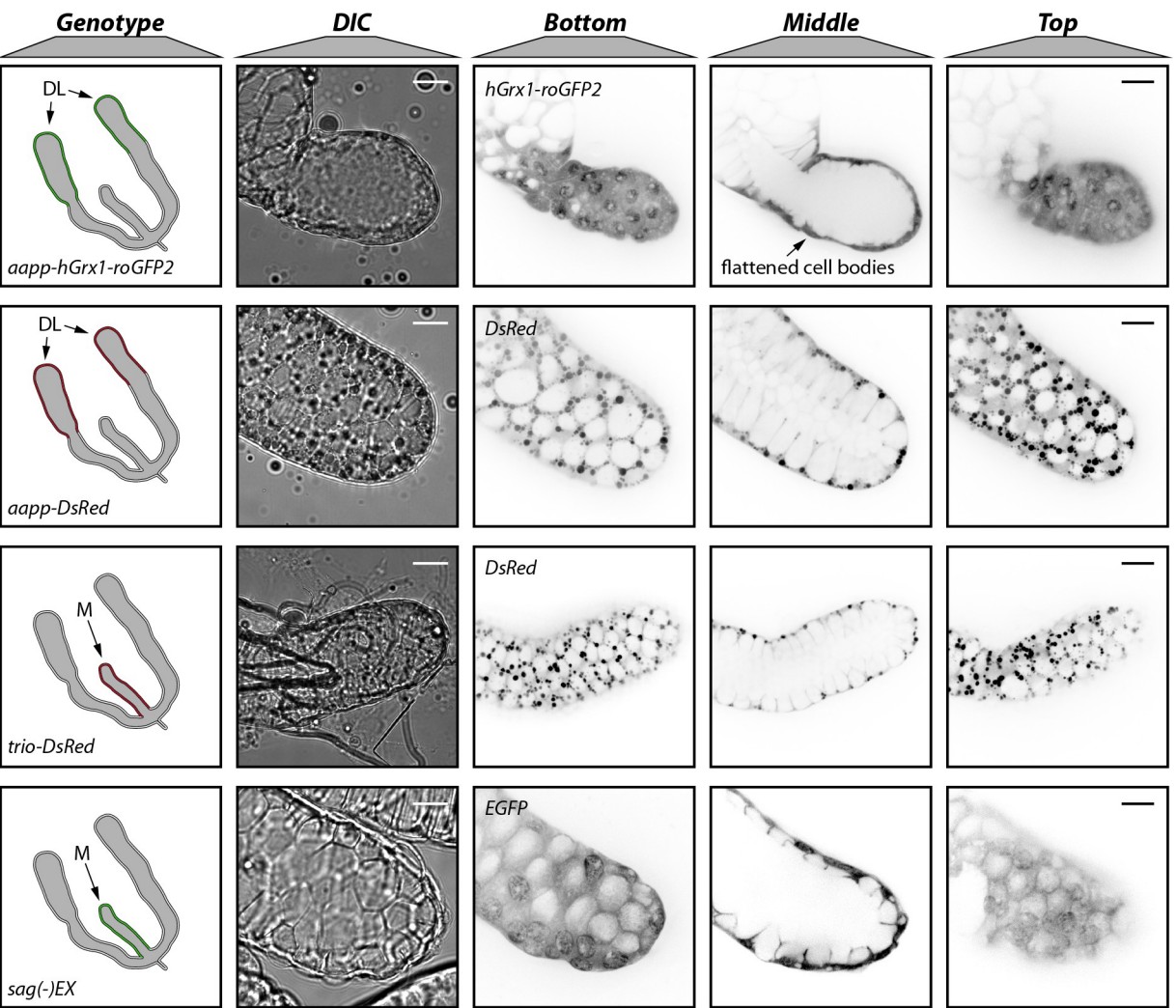

**Fig 4. Different subcellular localizations of DsRed, EGFP and hGrx1-roGFP2 fluorescent reporters.** Differential interference contrast (DIC) and confocal fluorescence images of the distal-lateral lobes (*aapp-DsRed*, *trio-DsRed*) or the median (*trio-DsRed*, *sag(-)EX*) salivary gland lobes. Different z-sections (top, middle and bottom) of the same glands are displayed in the relevant fluorescence channel. Cell bodies of acinar cells in the tips of distal lobes expressing high levels of hGrx1-roGFP2 appeared flattened (as indicated) compared to cells expressing *aapp-DsRed*. Note that DIC and fluorescence images were taken with a Hamamatsu Orca Flash 4.0 V1 and a Yokogawa CSU X-1 camera, respectively. Images between different channels are therefore not completely matching although depicting the same salivary gland. Illustrations on the left depict the genotype of the lines and corresponding expression patterns. All images were acquired in the same microscopy session. Scale bars: 20 μm.

were observed exclusively in male pupae (**S6B Fig**). In contrast, the presence of a fluorescent ocellus or body fluorescence was sex-independent. Quantification of the observed expression pattern revealed that a fluorescent ocellus, either alone or in combination with body fluorescence, was most frequently observed, and that the ratio of pupae that displayed body fluorescence to those that did not was close to 50% (**S6C Fig**). To follow the inheritance pattern of high and low body fluorescence, pupae were separated according to their fluorescence phenotype and intercrossed. Quantification of pupal *DsRed* expression in the F1 generation revealed a strong enrichment for whole body fluorescence (~80%) if high pupal body fluorescence was selected in the previous generation. In contrast, no pupae with high body fluorescence were observed in the offspring of mosquitoes hatched from low fluorescent pupae (**S6D Fig**). This result is consistent with Mendelian inheritance, with the locus conferring high body

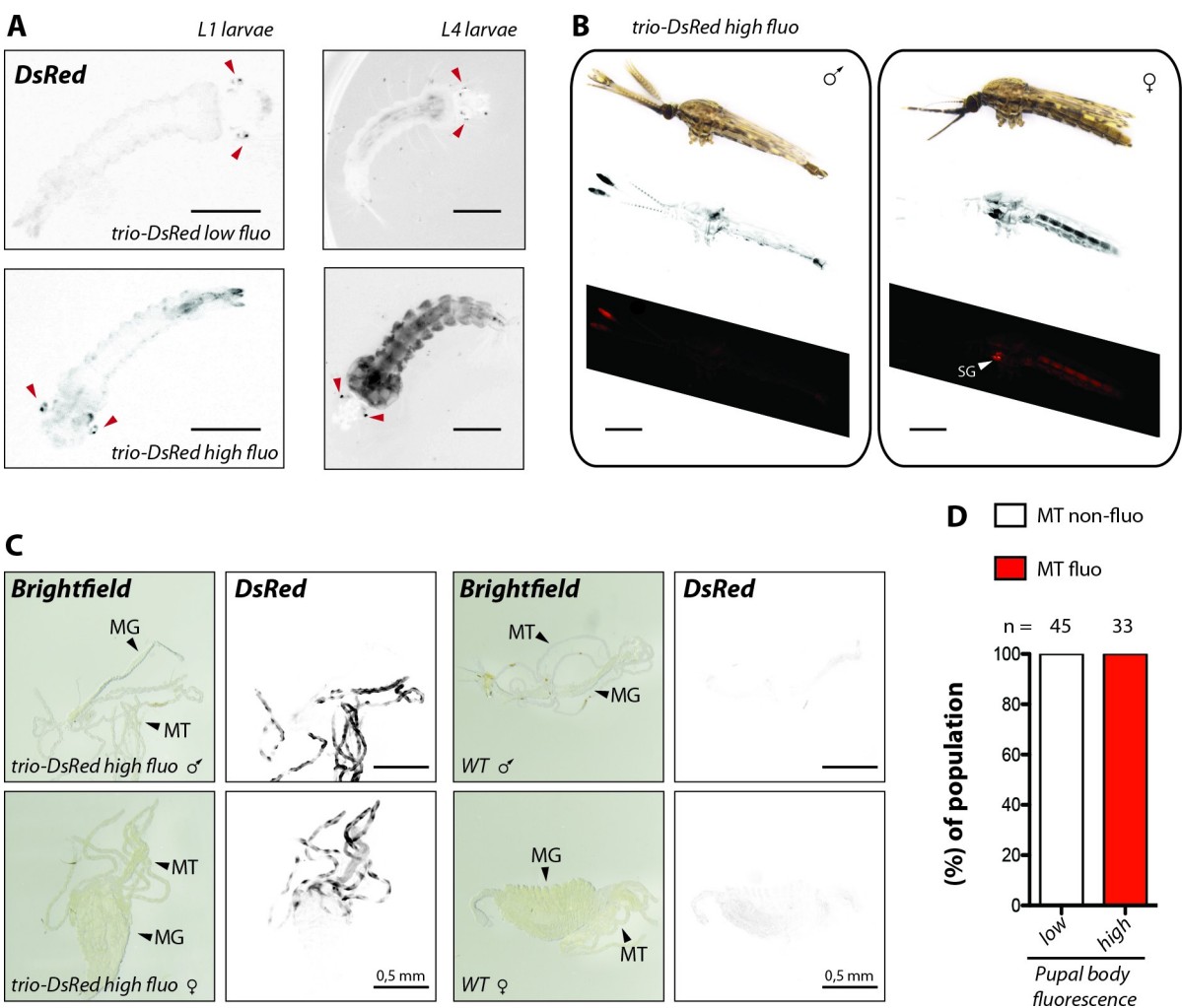

**Fig 5.** *DsRed* **expression in Malphigian tubules (MT) of** *trio-DsRed* **mosquitoes is linked to ubiquitous** *DsRed* **expression in larvae, pupae and adults. A)** *Trio-DsRed* L1 and L4 larvae were classified as *trio-DsRed low fluo* or *trio-DsRed high fluo* according to their body DsRed fluorescence intensity. Red arrowheads point towards the DsRed positive eyes observed in the majority of larvae. Scale bar L1: 250 µm. Scale bar L4: 0.5 mm. **B)** Whole body fluorescence of *trio-DsRed high fluo* male and female mosquitoes. From top to bottom: brightfield, inverted DsRed fluorescence signal displayed in black and white and in red on black. Note that images were stitched from two separate images showing either the head or the abdominal part. Scale bar: 0.5 mm. **C)** Representative images of male and female midguts dissected from mosquitoes that displayed high body fluorescence at the pupal stage in comparison to *wild type* (WT: *G3*). Malphigian tubules (MT) and midguts (MG). Scale bar: 0.5 mm. **D)** Correlation of high body fluorescence at the pupal stage with *DsRed* expression in malphigian tubules (MT). Data were collected from two different mosquito generations. The number of analyzed mosquitoes is indicated above each column.

fluorescence being dominant over that conferring low body fluorescence. To investigate this further, virgin female pupae of the F1 generation displaying high pupal *DsRed* expression were crossed with wild type males. Ten single female families were created which were all screened for pupal body fluorescence. All families except family 8 displayed a 1:1 distribution for high and low pupal body fluorescence (**S6E Fig**). Family 8 displayed exclusively high pupal body fluorescence, indicating that the founder female was homozygous for the locus conferring high body fluorescence, while all other females were heterozygous for the „low"and the „high"*trio-DsRed* loci. Notably, we detected high DsRed signals in the salivary glands of all adult females in families 5, 6, 7, 8 and 10 (other families were not investigated), regardless of whether individuals hatched from pupae displaying high or low body fluorescence. Genotyping of females

homozygous for low *trio-DsRed* and heterozygous for WT/high *trio-DsRed* revealed a single integration of the transgene into the *X1* locus as expected (**S7 Fig**).

## Glutathione redox state and morphology of acinar cells expressing hGrx1-roGFP2

To assess the redox sensitivity of hGrx1-roGFP2 expressed in the *aapp-hGrx1-roGFP2* line, salivary glands were dissected and treated either with diamide or dithiothreitol to oxidize or reduce the fluorescent probe, respectively. In addition, salivary glands were treated with N-ethyl-maleimide to preserve the native redox state of the dissected tissue. Samples were imaged at 405 and 488 nm and the ratio of both wavelengths was plotted (**Fig 6A**), revealing a dynamic range of 4.2 between fully oxidized and fully reduced probes (**Fig 6B**). Interestingly, the ratio of native salivary glands was very similar to measurements performed on samples treated with dithiothreitol (Red), suggesting that the glutathione pool is largely reduced in acinar cells (**Fig 6B**). Infection with *P. berghei* sporozoites did not affect the redox state of salivary glands (**Fig 6C**). Of note, initial imaging experiments pointed towards a thinning of the tips of the distal-lateral lobes in salivary glands expressing *aapp-hGrx1-roGFP2*, while glands obtained from *aapp-DsRed*, *trio-DsRed* and *sag(-)KI* or *sag(-)EX* females appeared similar to wild type (**Figs 1 and 6D**). Measurement of the diameter at the "neck-region" of the distal-lateral lobes confirmed its thinning in salivary glands expressing *aapp-hGrx1-roGFP2* compared to wild type and *aapp-DsRed* (**Fig 6E**). This difference is likely due to a flattening of the cell body of the acinar cells within the tips, which was observed by confocal microscopy (**Fig 4**).

## Salivary gland reporter lines as a tool to investigate sporozoite interactions with salivary glands

We next investigated whether our highly fluorescent salivary gland reporter lines could facilitate *in vivo* imaging of *Plasmodium* development through the cuticle of a living mosquito using a *GFP* expressing parasite strain (*Δp230p-GFP*) of the rodent malaria parasite *P. berghei* (**Fig 7A**) [23]. This strain is bright enough to make parasites visible through the mosquito cuticle, allowing pre-sorting of infected mosquitoes based on fluorescence in the mosquito midgut, the salivary glands and the wing joint, a localization where hemolymph sporozoites tend to accumulate (**Fig 7B**). To test if the expression of a fluorescent reporter in the salivary glands alters parasite development in the mosquito, we infected the three reporter lines *trio-DsRed*, *aapp-DsRed* and *aapp-hGrx1-roGFP2*, as well as the parental line *G3*. Parasite counts observed in reporter lines varied from 118 to 142 oocysts and were not significantly different from controls (121 oocysts) (**S8A Fig**). The average number of salivary gland sporozoites at day 17–18 after infection was 8.200 to 17.400 sporozoites per mosquito in salivary gland reporter lines vs 9.600 in *G3* controls (**S8B Fig**). An average of 10.000 salivary gland sporozoites per mosquito is usually observed in our laboratory, with numbers fluctuating from 5.000 to more than 20.000 sporozoites per mosquito in non-transgenic lines, indicating no significant difference between the *G3* control and reporter lines. To normalize the counts of salivary gland sporozoites relative to differences in oocyst numbers, we calculated the number of sporozoites that had successfully invaded the salivary glands per oocyst. Again, no significant difference was detected between reporter lines (86–146 sporozoites per oocyst) and *G3* control (133) mosquitoes (**S8C Fig**).

While the expression of a fluorescence reporter makes it simple to detect the location of the salivary glands through the cuticle, light absorption by pigments and light scattering by chitin generate signal losses reducing image quality and resolution. To decrease light absorption, we crossed *aapp-DsRed* mosquitoes, displaying highly fluorescent salivary glands without altered

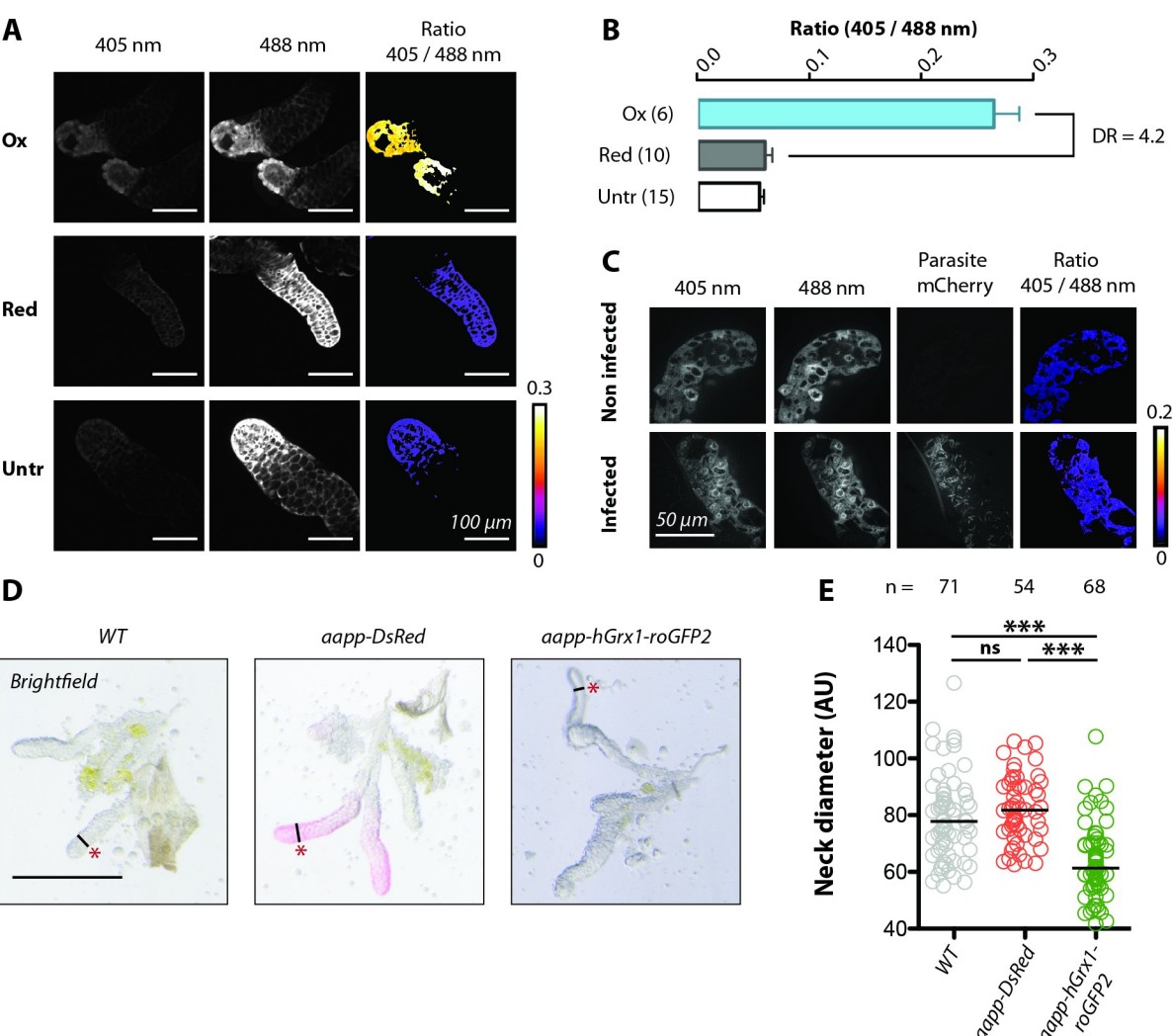

**Fig 6. hGrx1-roGFP2 expressed in the salivary glands is functional but affects gland morphology. A)** Dissected salivary glands expressing *hGrx1-roGFP2* were incubated in diamide (ox), dithiothreitol (red) or N-ethylmaleimide (untr) and imaged at 405 nm and 488 nm wavelengths. The 405/488 ratio is given in false colors to indicate the oxidation level of the glutathione pool. Scale bars: 100 μm. **B)** Quantification of glutathione oxidation based on the 405/488 ratio. Salivary glands were treated as in (A). Numbers in parentheses indicate the number of examined salivary glands. DR: dynamic range. **C)** Image of a salivary gland infected with *mCherry*-expressing *P. berghei* sporozoites. Staining and imaging were performed on two consecutive days with ≥10 salivary glands per experiment. A representative image is shown. Scale bar: 50 μm. **D)** Brightfield images of salivary glands from wild type (*G3*), *aapp-DsRed* and *aapp-hGrx1-roGFP2* females. The „neck region" of the distal-lateral lobes is indicated by a red asterisk and the measured diameter by a black line. Between 54 and 71 images were acquired for each line in ≥3 independent preparation and imaging sessions. Representative images presented here were all acquired in the same session. Scale bars: 250 μm. **E)** Diameter measurements of the distal-lateral lobe neck region of *aapp-DsRed* and *aapp-hGrx1-roGFP2* in comparison to wild type (*G3*). The number of measurements is indicated above each genotype. Note that only the diameter of one distal-lateral lobe per gland was measured. Data represent ≥3 independent experiments. The black line indicates the median. ***p<0.0001, one-way-ANOVA (Bonferroni's Multiple Comparison Test). ns: not significant.

morphology, with *yellow(-)KI* mosquitoes presenting a pale and yellowish cuticle (**Fig 7A and 7B**). To assess if the *3xP3* driven expression of *EGFP* in the low pigmented *yellow(-)KI* line would interfere with the imaging of GFP positive *P. berghei* parasites, adult *yellow(-)KI* and WT females were imaged in the GFP channel to compare emitted fluorescence signals (**S8D Fig**). *Yellow(-)KI* females showed a weak GFP signal in the head and, surprisingly, strongly fluorescent ovaries (**S8D Fig**). While gene expression within parts of the eye and the nervous system of the head is typical for the expression pattern of the *3xP3* promoter in mosquitoes,

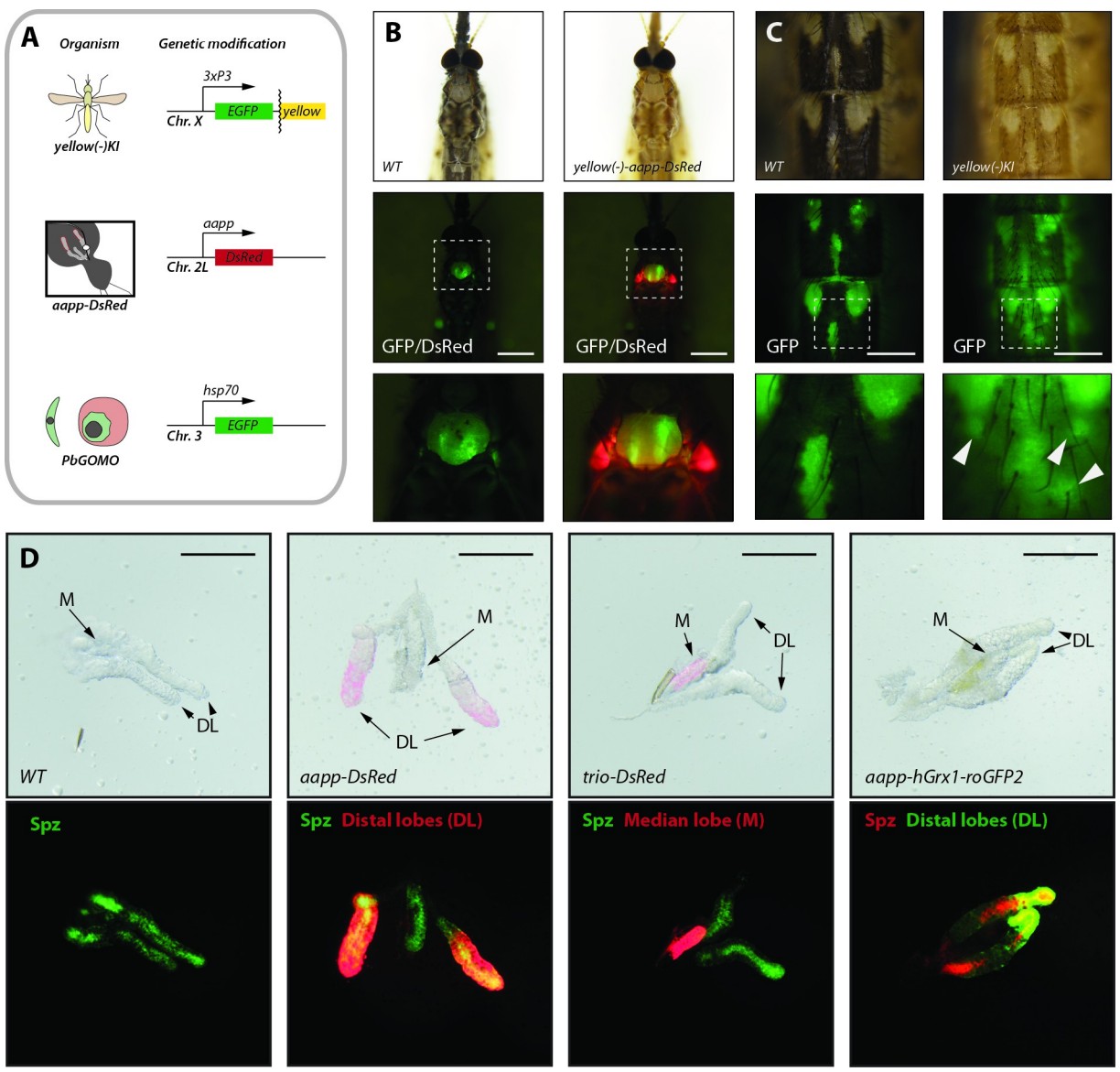

**Fig 7. *Yellow(-)* and salivary gland reporter lines as tools for *in vivo* imaging. A)** Genetic models used for *in vivo* imaging. Mosquitoes carrying the *aapp-DsRed* transgene were crossed with *yellow(-)KI* mosquitoes displaying reduced black pigmentation. Infections were carried out using the highly fluorescent *P. berghei* line *Δp230p-GFP* constitutively expressing *EGFP*. **B)** Pigmentation phenotype and salivary gland-specific *DsRed* expression in *P. berghei* infected *yellow(-)-aapp-DsRed* and wild type (*WT*) mosquitoes. *GFP*-expressing sporozoites are displayed in green. Brightfield and DsRed signal of the same field of view. Note that the green dots in the lower thorax area represent hemolymph sporozoites trapped in the wing joints of mosquitoes. The images below show enlargements of the framed areas in the images above. Scale bar: 0.5 mm. **C)** Images of the abdomen of infected *yellow(-)KI* and wild type (*Ngousso*) mosquitoes. Brightfield and GFP expression of the same field of view. The images below show enlargements of the framed areas in the images above. Note the better light transmission in the abdomen of the *yellow(-)KI* mosquito. White arrows point towards oocysts visible through more pigmented areas of the cuticle of *yellow(-)KI* mosquitoes. Scale bar: 250 μm. **D)** Images of salivary glands highly infected with *P. berghei*. Wild type (*WT*), *aapp-DsRed* and *trio-DsRed* have been infected with *GFP* expressing *P. berghei* (*Δp230p-GFP*) while *aapp-hGrx1-roGFP2* mosquitoes were infected with *mCherry* expressing *P. berghei* (*Δp230p-mCherry*). M = median lobe; DL = distal lobes. Scale bar: 250 μm.

EGFP expression in the ovaries is likely due to the endogenous promoter activity of the *yellow* promoter. However, no significant increase in green fluorescence was observed at the throat area where the salivary glands localize compared with WT, suggesting that imaging of GFP-positive parasites would be feasible.

As expected, *yellow(-)-aapp-DsRed* mosquitoes displayed a low pigmented cuticle in combination with highly fluorescent salivary glands (**Fig 7B**). The decrease in pigmentation greatly increased the observed GFP signal visible through the cuticle, notably in strongly pigmented areas that appear almost black in WT mosquitoes (**Fig 7C**). We also investigated whether GFP- or mCherry-positive sporozoites can be imaged in fluorescent salivary glands. Imaging of dissected infected glands showed that fluorescent sporozoites can be clearly distinguished from the fluorescent signal of salivary glands at low magnification while lobe-specific fluorescence expression allowed for simple differentiation between median and distal-lateral lobes (**Fig 7D**). In addition, imaging showed that all salivary gland reporter lines may have a high sporozoite load, suggesting that fluorescence expression in salivary glands does not affect individual salivary gland sporozoite colonization.

We next imaged mosquitoes infected with *P. berghei Δp230p-GFP* between day 17 and day 20 after infection, when the majority of sporozoites have invaded the salivary glands. Mosquitoes were pre-sorted for GFP fluorescence in the salivary glands (**S9A Fig**) as well as in the wing joint (**Fig 7B**) to confirm the presence of sporozoites. In a subset of mosquitoes (less than 1 out of 5), we observed that the salivary glands localized very close to the cuticle (**S9B and S9C Fig**). To facilitate *in vivo* imaging, we further selected mosquitoes positive for sporozoites and with cuticular localization of at least one salivary gland and prepared them for imaging (**S9D Fig**). Confocal imaging enabled to visualize precisely single sporozoites through the cuticle close to the salivary gland as well as subcellular structures, like the cavities of acinar cells of the gland itself (**Fig 8A**). However, we noted that a significant portion of the DsRed signal was also seen in the GFP channel, especially at lower magnification, suggesting a possible spillover of the DsRed signal. This could possibly occur due to the concentration difference between DsRed in the acinar cells and EGFP in the sporozoites (**S9E Fig**). Nevertheless, spillover effects did not affect detection of sporozoites at higher magnification. Quantification of the recorded movies revealed that ~26% of persistent sporozoites in the thorax area (at least 5 minutes in the field of view) perform pattern of active movement while ~19% are attached but not moving and ~55% are floating (**Fig 8A**). The observed movement patterns resembled in most cases the patch gliding previously described *in vitro*. During this type of movement, the sporozoite adheres in the center and performs jerky back-and-forth movements along its adhesive patch (**Fig 8B and** S1 and S2 **Movies**) [24,25]. Due to the brevity of the recorded movies (5 minutes), many actively moving sporozoites appeared to remain on the site without making large terrain gains. However, in some cases, the position of the adhesive patch was observed to change slightly over time, such that sporozoites made small terrain gains in one direction while continuously sliding back and forth. Importantly, we were also able to observe sporozoites inside salivary glands (**Fig 8C**).

## Discussion

Mosquito salivary glands have long interested entomologists, parasitologists, virologists and microbiologists because they are a key organ in the transmission of vector borne diseases. Still, the host-pathogen interactions that allow the specific recognition and colonization of this organ by different pathogens has been understudied, probably in part because of the relative difficulty to access and visualize these processes in comparison to midgut invasion. The reporter lines we generated in this study represent valuable tools to circumvent some of these issues: they enable [1] easy visualization of the glands, even in brightfield or DIC microscopy, [2] distinguishing distal-lateral and median lobes of the salivary glands, and [3] imaging of single sporozoites in proximity to and inside salivary glands *in vivo*.

We used the promoters of three salivary gland genes, *aapp*, *trio* and *saglin*, to drive fluorescent reporter expression in *A. coluzzii* mosquitoes. The different fluorescence patterns

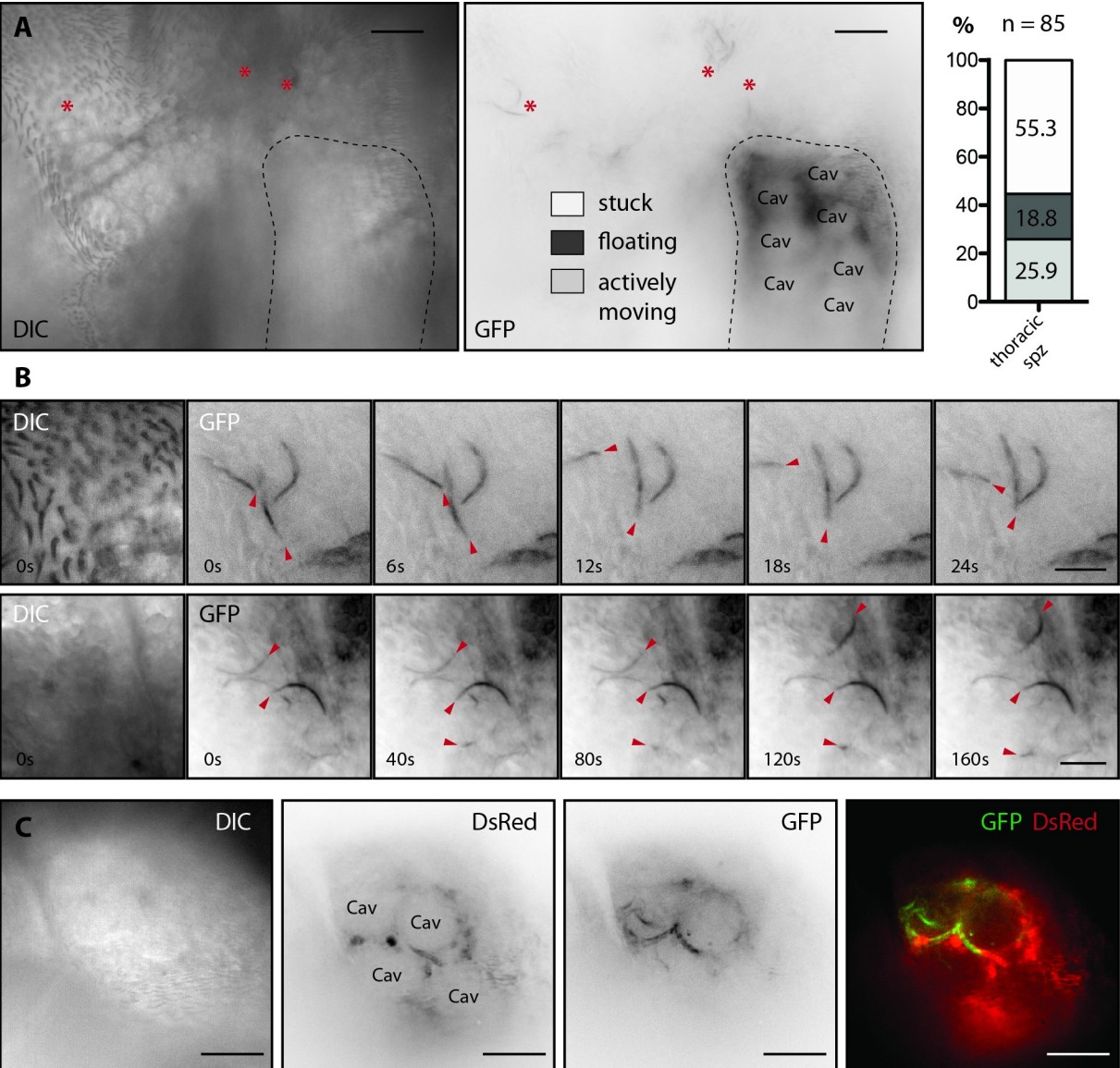

**Fig 8. Confocal imaging of sporozoite motility in live mosquitoes. A)** View of the thorax area of a *yellow(-)-aapp-DsRed* female mosquito with persistent sporozoites indicated by red asterisks in the proximity of the salivary gland (outlined with a dashed line). Differential interference contrast image (DIC, left) and GFP signal (right). Note that *DsRed* expression in the salivary glands was strong enough to be visible in the GFP channel. Cav: salivary gland cavity. Scale bar: 20 μm. The behavior of observed persistent sporozoites was classified as "stuck", "floating" and "actively moving". The number of examined sporozoites is indicated above the column. Combined results of four different experiments. **B)** Time series of sporozoites showing active movements. DIC images at 0 s and time series of the GFP signal with 6 s (top) and 40 s (bottom) between frames. Red arrows point towards sporozoites displaying active movement. Scale bar: 10 μm. **C)** Sporozoites inside the salivary gland imaged through the cuticle of a living mosquito. Left to right: DIC, inverted signals of DsRed and EGFP in black on white and merge of both in colors. Note that the sporozoites follow the circular shape of the cavities. Scale bar: 20 μm.

obtained in the salivary glands of these lines show that the proteomes of median and distal-lateral lobes are diverse and most likely contribute different saliva components. In addition, the variable fluorescence intensity in the distal-lateral lobes of both *aapp* lines (stronger towards the apex) point towards an even more complex expression pattern along the lobe. This hierarchical expression was even more evident for the *trio* and *saglin* promoters, both of which are active in the median lobe but with opposite expression patterns. *Saglin* is strongly active in

cells at the base and less active in the apex, in contrast, *trio* expression is absent in cells with high *saglin* activity, but very strong in cells with low *saglin* activity. While the expression pattern of *trio* has not been investigated before, the observed activity of the *saglin* promoter is coherent with immunofluorescence studies performed in *A. stephensi* [15]. Expression of *saglin* in the median lobe is puzzling since the protein has been shown to affect colonization of the salivary glands by *Plasmodium* sporozoites, [13], whereas parasites prefer to invade the distal-lateral lobes [5] that lack activity of the *saglin* promoter. A possible explanation is that its site of expression is not the site where it interacts with parasites.

The absence of reporter expression in mosquitoes expressing *DsRed* controlled by the *saglin* 5' upstream sequence is likely due to the shortness of the cloned sequence. The *saglin* gene localizes in the middle of an array of four genes believed to be salivary gland-specific [16] and separated by short intergenic regions (**S2 Fig**). Regulatory sequences controlling transcription are likely located upstream and/or downstream of this array and might be shared by all four genes.

Notably, the reporter lines display different subcellular localizations of hGrx1-roGFP2, EGFP and DsRed. While all expressed fluorescent proteins lack any targeting signal which could explain a different localization, only hGrx1-roGFP2 (*aapp*) and EGFP (*saglin*) localized within the cytoplasm. In contrast, DsRed displayed a vesicular pattern in both the median (*trio*) and the distal-lateral lobes (*aapp*), while a DsRed2 variant driven by the *aapp* promoter in *A. stephensi* mosquitoes was shown to be cytoplasmic [12]. Although AAPP was shown previously to localize in a vesicle-like fashion [6], we believe that the observed localization difference is likely artefactual due to fortuitous trafficking signals present in the reporter proteins. First, vesicular localization of DsRed was observed with both the *trio* and *aapp* promoters, and second, only DsRed appeared vesicular, whereas hGrx1-roGFP2 was cytoplasmically localized, although both proteins were expressed from the same *aapp* sequence. Comparison of the DsRed sequence used here with the previously described DsRed2 sequence expressed in *A. stephensi* revealed mutations mainly in the N-terminal region and the addition of 11 unrelated amino acids at the C-terminus of DsRed2 [12]. It is likely that these sequence differences between the two DsRed versions lead to the formation of a cryptic trafficking signal recognized by the acinar cells. Interestingly, although we did not investigate their nature, we also observed the presence of vesicles in the salivary glands of wild type mosquitoes with similar distribution and morphology and displaying weak red autofluorescence (**S2 Fig**). DsRed could potentially be trafficked to these or similar compartments explaining its unique localization. Differences in salivary gland morphology between *aapp-DsRed* and *aapp-hGrx1-roGFP2* expressing mosquitoes are likely due to the expression of human Glutaredoxin 1 (hGrx1) and not of roGFP2, a protein derived from EGFP with 2 amino acid changes. Indeed, no change in morphology was detected in *sag(-)EX* and *sag(-)KI* mosquitoes expressing EGFP alone in the median lobe. Still, EGFP appeared less expressed in *sag(-)EX* compared to hGrx1-roGFP2 in *aapp-hGrx1-roGFP2* (**Fig 1**) and we cannot exclude that the acinar cells of the median and distal-lateral lobes react differently to the expression of the respective reporters, or that the cells show different responses to roGFP2 and EGFP. Despite its influence on cell morphology, the salivary glands of hGrx1-roGFP2 expressing mosquitoes retained their integrity even in old mosquitoes and apoptosis of cells expressing the fluorescent reporter was not observed.

We also report a surprisingly large panel of fluorescence patterns in mosquitoes expressing the *trio-DsRed* transgene. The *trio* promoter activated transcription of *DsRed* in the salivary glands of all females (but not in males), but also in ocelli, antennae and Malpighian tubules as well as whole body in some but not all mosquitoes, independently of their sex but for antennae that were strictly restricted to males, and with a strict correlation between strong *DsRed* expression in pupal body and adult Malpighian tubules. The fact that *trio* expression was

reported to be salivary gland-specific [17] suggests that the region we selected upstream of *trio* does not contain all regulatory elements that control its natural expression, or that the genomic context of the *X1* locus may influence its expression. To date, the number of available promoters able to drive ubiquitous expression of transgenes in *Anopheles* mosquitoes is limited [26], thus replacing *DsRed* in a selected highly fluorescent line could provide a new means to ubiquitously express transgenes. By sorting and crossing according to the observed pupal fluorescence phenotype, it was possible to separate alleles conferring high and low pupal body fluorescence, which enabled the production of a colony enriched with the high pupal fluorescence phenotype and a colony showing only low pupal body fluorescence. This observation pointed towards a genetically encoded difference which is inherited in a Mendelian manner rather than a transient (epi)genetic modification. Genotyping confirmed the presence of a single copy of the transgene in the *X1* locus in both weakly and highly fluorescent mosquitoes. Since fluorescent ocelli, antennae and whole-body expression were only observed in the *trio-DsRed* population, we propose that the different *DsRed* expression patterns are locus-specific variegation effects caused by interactions between the selected *trio* promoter sequence and polymorphic regulatory elements that could be linked or unlinked to the transgene. Although unlikely, we cannot rule out the possibility that some mosquitoes carried an extra copy of the transgene that caused the highly fluorescent phenotype. Still, DsRed positive ocelli and antennae were observed in both high and low fluorescence populations, which is most likely explained by a variable *DsRed* expression from the transgene inserted at *X1*. While the observed *DsRed* expression patterns in larvae and pupae are helpful in the laboratory to select transgenic individuals, it also urges caution in interpreting results as it clearly shows that transcription is contextual.

Intravital imaging has revealed the dynamics of parasite development inside its vertebrate host [27,28]. While optical windows in mice is a well implemented method [29], *in vivo* imaging through the cuticle of insects is rarely performed, mostly because it is difficult to achieve good resolution by imaging through the insect cuticle that scatters and absorbs light. Recent studies have explored the possibility to circumvent this issue through the application of clearing protocols on infected mosquitoes to improve resolution and enable imaging through whole mosquitoes [30,31]. However, while these improvements are remarkable, imaging of cleared specimens only represents snapshots whereas mapping of dynamic processes are desirable. Imaging of tissues *ex vivo* in combination with the use of hemolymph-like media allows the observation of dynamic processes [32]. Still, imaging *ex vivo* samples can only be performed for a limited amount of time and artifacts due to the artificial tissue environment cannot be excluded. A step towards intravital imaging was made recently by developing a protocol that combines feeding of dyes relevant for imaging with blood feeding prior to sample preparation [33]. Due to the inflation of the mosquito abdomen directly after feeding, cuticle plates open up and create temporary optical windows that can be exploited for microscopy. Still, in this case, mosquitoes were decapitated and tightly flattened between microscopy slide and cover slip, thus enabling optimal imaging but maintaining *in vivo* conditions only for a limited time. To demonstrate the use of the salivary gland reporter lines for imaging purposes, we crossed *aapp-DsRed* mosquitoes with the weakly pigmented mosquito line *yellow(-)KI* deficient for black pigment synthesis. In a subset of mosquitoes, salivary glands localized very closely to the cuticle allowing the recognition of subcellular structures of the gland in living mosquitoes. In addition, the more transparent cuticle enabled better visibility of individual sporozoites through the cuticle. It is generally believed that sporozoites, after they have been released from the oocyst, are passively transported to the salivary glands and engage in active motility only upon contact with acinar cells [3]. Our *in vivo* observations show that ~26% of persistent thorax sporozoites perform active motility mostly in a patch gliding manner (**S1** and

S2 **Movies**) [24,25]. This could be an indication that the motility of the sporozoites is already crucial to reach the salivary glands and ensure efficient colonization. Testing mutants that produce immobile sporozoites could provide more sophisticated information on the extent to which sporozoite motility promotes invasion of salivary glands.

Here we provide a proof of concept of *in vivo* salivary gland imaging, and we hope that the method can be further improved to gain a better insight in the invasion process of salivary glands by sporozoites. Potential spillover effects of the DsRed signal in salivary glands of *aapp-DsRed* females could possibly be avoided by using a *P. berghei* reporter line with stronger GFP expression [34]. Furthermore, the injection of fluorescent beads or dyes could be used as vitality measure to monitor the active movement of the hemolymph while, at the same time, highlighting contact areas with hemolymph and salivary glands. The localization of most sporozoites, separated from the objective by several cell layers, was a limiting factor in our settings, which could be improved by using two-photon microscopy instead of confocal microscopy, thus providing a better resolution when imaging in deep tissues. Recently, this method has been used for long-term monitoring of the ventral nerve cord in live adult *Drosophila*, although the cuticle was previously partially removed for this approach [35]. Sample selection and preparation, like partial embedding of mosquitoes as well as the choice of the immersion medium, could also lead to significant improvements. Furthermore, we did not exhaust the repertoire of possible genetic modifications to decrease pigmentation of the cuticle. Deletion of the genes *tan* and *ebony*, two enzymes which regulate the homeostasis of the pigment precursors dopamine and N-ß-alanyldopamine, in combination with the deletion of *yellow* could potentially further decrease cuticle pigmentation [21,36]. Another possibility could be the overexpression of arylalkylamine-*N*-acetyltransferase, an enzyme converting dopamine into the sclerotizing precursor molecule N-acetyldopamine, which was shown to affect pigmentation in different insect species by enhancing the production of colorless components of the cuticle [37]. A key improvement for *in vivo* imaging would be an increase in the number of mosquitoes displaying ideally positioned salivary glands. Preliminary monitoring experiments showed that the position of the gland does not significantly change over time in adult mosquitoes indicating that positioning occurs during metamorphosis. Should salivary gland positioning be genetically determined, selective breeding may help to improve the ratio of mosquitoes with cuticular salivary glands. Taken together, we hope that the transgenic lines developed in this study can contribute to improve *in vivo* imaging and salivary gland dissections in *Anopheles* mosquitoes, and enable real-time imaging of *Plasmodium* and other vector borne pathogens.

## Materials & methods

### Ethics statement

Experiments were carried out in conformity with the 2010/63/EU directive of the European Parliament on the protection of animals used for scientific purposes. Our animal care facility received agreement #I-67-482-2 from the veterinary services of the département du Bas Rhin (Direction Départementale de la Protection des Populations). The use of animals for this project was authorized by the French ministry of higher education, research and innovation under the number APAFIS#20562–2019050313288887 v3. The generation and use of transgenic lines (bacteria, mosquitoes, parasite) was authorized by the French ministry of higher education, research and innovation under the number 3243.

### Animals

Transgenesis was performed in the *Anopheles coluzzii* strain X1 [19] or in a strain expressing Cas9 from the *X1* locus under control of the *vasa* promoter (transgenesis plasmid: Addgene #

173670) and introgressed into the *Ngousso* genetic background. Wild type *Anopheles coluzzii* strains *Ngousso* and *G3* were used as controls. CD1 mice purchased from Charles River or from our own colonies were used for both mosquito maintenance through blood meals, and for parasite maintenance and mosquito infections. For infections with rodent malaria parasites, we made use of the *Plasmodium berghei* lines *Δp230p-GFP* and *Δp230p-mCherry* derived from the wild type background *P. berghei ANKA* [41].

## Breeding of mosquitoes

*Anopheles coluzzii* mosquitoes were kept in standard conditions (27˚C, 75–80% humidity, 12-hr/12-hr light/dark cycle). Larvae were hatched in pans of osmosed water and fed with pulverized fish food (TetraMin, Melle, Germany). Adult mosquitoes were maintained on a 10% sucrose solution. To propagate colonies, four to seven-day old mosquitoes were blood fed for 10–15 minutes on anesthetized mice and allowed to lay eggs on wet filter paper 2–3 days later. For mosquito blood feeding, female CD-1 mice (> 35g) were anesthetized with a mixture of Zoletil (42.5 mg/kg) and Rompun (8.5 mg/kg) in 0.9% NaCl solution.

## Construction of salivary gland reporter transgenes

The upstream sequences of *saglin* (primers P9 and P10), *aapp* (primers P1 and P2) and *trio* (primers P7 and P8) were amplified from genomic DNA of *G3* mosquitoes isolated with the Blood & Tissue Kit (Qiagen) (**S1 Appendix**). *DsRed* (primers P5 and P6 for *aapp/trio*; P11 and P12 for *saglin*) and *hGrx1-roGFP2* (primers P3 and P4) were amplified from the plasmids *pJET-DsRed-SV40* and *pDSAP_actin-hGrx1-roGFP2* (donated by Raquel Mela-Lopez). Primers are described in **Table 1**. PCR was performed using Phusion polymerase according to the manufacturer recommendations (ThermoFisher Scientific) and subcloned in the *pJET* vector using the CloneJET PCR Cloning Kit (ThermoFisher Scientific). The upstream sequence of the *saglin* gene was fused to the *DsRed* gene by a Phusion mediated assembly PCR [42] before cloning into *pJET*. After sequence verification, assembly blocks were fused into the destination vector *pDSAP* (19) (Addgene # 62293) using Goldengate cloning and generating *pDSAP-aapp-DsRed*, *pDSAP-aapp-hGrx1-roGFP2*, *pDSAP-trio-DsRed* and *pDSAP-saglin-DsRed*.

## Construct cloning for site-directed mutagenesis into *yellow* and *saglin* genes

For the design of guide RNAs (gRNAs), the coding sequence of *saglin* as well as the first exon of the *yellow* gene were manually searched for PAMs (NGG), with N representing any nucleotide. Candidate gRNA sequences were investigated for potential off-target effects using http://www.rgenome.net/cas-designer/ (Cas-OFFinder) [43]. We selected gRNAs starting with a G considering the transcription start from the U6 promoter and differing by ≥3 nucleotides from putative off-target sequences in the *A. coluzzii* genome. For each knockin construct, linkers encoding the protospacer motif of three gRNAs (**Table 1**) were generated by annealing two primers ordered from IDT DNA and cloned using *BbsI* into *pKSB-sgRNA* (Addgene #173671–3) vectors containing the *U6 snRNA polymerase III* promoter (AGAP013557), the invariable sequence of a standard *crRNA-tracerRNA* fusion [44], and the *RNA polIII* TTTTT terminator. The three complete gRNAs blocks were then assembled into a single *pKSB-sgRNA* destination plasmid by GoldenGate cloning using different but matching *BsaI* overhangs. For the construction of repair templates, regions of homology on the 5' side (*saglin*: primers P29 and P30; *yellow*: primers P19 and P20) and on the 3' side (*saglin*: primers P31 and P32; *yellow*: primers P21 and P22) of the region targeted by the gRNAs were amplified from *Ngousso* genomic DNA and subcloned in the *pJET* vector, generating plasmids *pJET-5'saglin*, *pJET-5'yellow*,

**Table 1. Key Resources.**

| Reagent type (species) or resource | Designation | Source or reference | Identifiers | Additional information |
|---|---|---|---|---|
| strain, (*Escherichia coli*) | *DH5α* | Home-made | / | Chemically competent cells |
| Strain, (*Anopheles coluzzii*) | *X1* | [19] | / | *attP* docking line |
| Strain, (*Anopheles coluzzii*) | *vasa-Cas9* | [38] | / | Strain with germline-specific *Cas9* expression |
| Strain, (*Anopheles coluzzii*) | *Ngousso* | MR4 | MRA-1279 | Wild type strain |
| Strain, (*Anopheles coluzzii*) | *G3* | MR4 | MRA-112 | Wild type strain |
| Strain, (*Anopheles coluzzii*) | *aapp-DsRed* | this study | / | *DsRed* expression in distal lobes |
| Strain, (*Anopheles coluzzii*) | *trio-DsRed* | this study | / | *DsRed* expression in median lobe |
| Strain, (*Anopheles coluzzii*) | *aapp-hGrx1-roGFP2* | this study | / | *hGrx1-roGFP2* expression in distal lobes |
| Strain, (*Anopheles coluzzii*) | *sag-DsRed* | this study | / | No *DsRed* expression |
| Strain, (*Anopheles coluzzii*) | *sag(-)KI* | this study | / | Knockin into *Saglin* expressing *EGFP* from *3xP3* promoter |
| Strain, (*Anopheles coluzzii*) | *sag(-)EX* | this study | / | Knockin into *Saglin* with excised *3xP3* promoter |
| Strain, (*Mus musculus*) | *CD-1* | Janvier Labs | / | self-bred |
| Strain, (*Plasmodium berghei*) | *Δp230p-GFP* | [23] | / | Constitutive expression of *GFP* |
| Strain, (*Plasmodium berghei*) | *Δp230p-mCherry* | [23] | / | Constitutive expression of *mCherry* |
| Recombinant DNA reagent | *pENTR R4 ATCC LacZ GCTT* | [19] | Addgene: #1773668 | / |
| Recombinant DNA reagent | *pDSAP* | [19] | Addgene: #62293 | / |
| Recombinant DNA reagent | P1 / P44, *aapp 5'UTR* forward | this study | Primer | GGTCTCGATCCCTTTTCTTTTACCCTTTGTAACACGCTAATAACG |
| Recombinant DNA reagent | P2, *aapp 5'UTR* reverse | this study | Primer | GGTCTCGCTCTGTCGTTTTTTTGTTGTTGTGGTATTATTTCACTG |
| Recombinant DNA reagent | P3 / P46, *hGrx1-roGFP2* forward | this study | Primer | GGTCTCGAGAGATGGCTCAAGAGTTTGTGAACTGCAAAATC |
| Recombinant DNA reagent | P4, *hGrx1-roGFP2* reverse | this study | Primer | GGTCTCGAAGCTTACTTGTACAGCTCGTCCATGCCG |
| Recombinant DNA reagent | P5, *DsRed aapp / trio* forward | this study | Primer | GGTCTCGAGAGATGGTGCGCTCCTCCAAGAACG |
| Recombinant DNA reagent | P6, *DsRed aapp / trio* reverse | this study | Primer | GGTCTCGAAGCCTACAGGAACAGGTGGTGGCGGC |
| Recombinant DNA reagent | P7 / P45, *trio 5'UTR* forward | this study | Primer | GGTCTCGATCCTTTTCTGAGGTGATCTTTCGAAAAGATCG |
| Recombinant DNA reagent | P8, *trio 5'UTR* reverse | this study | Primer | GGTCTCGCTCTGTAACTGGGAGCAGATTGATTTTATGG |
| Recombinant DNA reagent | P9 / P42, *saglin 5'UTR* forward | this study | Primer | GGTCTCGATCCCTTTCATGCGTACAATGTGTATGATCACTG |

(*Continued*)

**Table 1.** (Continued)

| Reagent type (species) or resource | Designation | Source or reference | Identifiers | Additional information |
|---|---|---|---|---|
| Recombinant DNA reagent | P10, *saglin 5'UTR* reverse | this study | Primer | CGTTCTTGGAGGAGCGCACCATTTATAAGTGAACCTTATGCAAACCTTCCCTG |
| Recombinant DNA reagent | P11 / P40, *DsRed saglin* forward | this study | Primer | CAGGGAAGGTTTGCATAAGGTTCACTTATAAATGGTGCGCTCCTCCAAGAACG |
| Recombinant DNA reagent | P12, *DsRed saglin* reverse | this study | Primer | GGTCTCGAAGCCTACAGGAACAGGTGGTGGCGG |
| Recombinant DNA reagent | P13, *Yellow* gRNA1 forward | this study | Oligonucleotide | CCTTGTAGCGTGCCTGGTGGCCGC |
| Recombinant DNA reagent | P14, *Yellow* gRNA1 reverse | this study | Oligonucleotide | AAACGCGGCCACCAGGCACGCTAC |
| Recombinant DNA reagent | P15, *Yellow* gRNA2 forward | this study | Oligonucleotide | CCTTGGCGACTACGTGCCCACGAA |
| Recombinant DNA reagent | P16, *Yellow* gRNA2 reverse | this study | Oligonucleotide | AAACTTCGTGGGCACGTAGTCGCC |
| Recombinant DNA reagent | P17, *Yellow* gRNA3 forward | this study | Oligonucleotide | CCTTGGAGAACAAGCTGTTCGTCT |
| Recombinant DNA reagent | P18, *Yellow* gRNA3 reverse | this study | Oligonucleotide | AAACAGACGAACAGCTTGTTCTCC |
| Recombinant DNA reagent | P19 / P33, *Yellow 5'UTR* forward | this study | Primer | GGTCTCGAACAGAATTGAACGCCCAAGGTACGGCTG |
| Recombinant DNA reagent | P20 / P34, *Yellow 5'UTR* reverse | this study | Primer | GGTCTCACCCCGCTTCTGCCACTAAGGAACTTCTGTG |
| Recombinant DNA reagent | P21, *Yellow 3'UTR* forward | this study | Primer | GGTCTCGAAGACGTGTGTCCTGGAAGTGTGATTTATCG |
| Recombinant DNA reagent | P22, *Yellow 3'UTR* reverse | this study | Primer | GGTCTCGAAGCGTAGCGTGCCTGGTGGCCGCCGGAACGCTTTCCCTGCAGCGTAC |
| Recombinant DNA reagent | P23, *Saglin gRNA1* forward | this study | Oligonucleotide | CCTTGCGTTCAAGTCATCTCGAGC |
| Recombinant DNA reagent | P24, *Saglin gRNA1* reverse | this study | Oligonucleotide | AAACGCTCGAGATGACTTGAACGC |
| Recombinant DNA reagent | P25, *Saglin gRNA2* forward | this study | Oligonucleotide | CCTTGCCGCAGTCCTCGCGGCTGG |
| Recombinant DNA reagent | P26, *Saglin gRNA2* reverse | this study | Oligonucleotide | AAACCCAGCCGCGAGGACTGCGGC |
| Recombinant DNA reagent | P27, *Saglin gRNA3* forward | this study | Oligonucleotide | CCTTGAAGTTTGCAACGTTTGCGA |
| Recombinant DNA reagent | P28, *Saglin gRNA3* reverse | this study | Oligonucleotide | AAACTCGCAAACGTTGCAAACTTC |
| Recombinant DNA reagent | P29, *Saglin 5'UTR* forward | this study | Primer | GGTCTCGAACATGGGTTCACTCGCTGTTAGACTGTG |
| Recombinant DNA reagent | P30, *Saglin 5'UTR* reverse | this study | Primer | GGTCTCAATGGTTATAAGTGAACCTTATGCAAACCTTCCCTG |
| Recombinant DNA reagent | P31, *Saglin 3'UTR* forward | this study | Primer | GGTCTCGAAGAGCGTCGGCGACGGTCTAAAAG |
| Recombinant DNA reagent | P32, *Saglin 3'UTR* reverse | this study | Primer | GGTCTCGAAGCGTTCAAGTCATCTCGAGCCGGCTTCTTCTGCTGCCGCAGAAAC |
| Recombinant DNA reagent | P35, *X1* forward 1 | this study | Primer | CCCGTCATGAGTTCAAGCCTGAA |
| Recombinant DNA reagent | P36, *attB* reverse | this study | Primer | CCCGGTCTCAAATTGCCCGCCGTGACCGTCGA |
| Recombinant DNA reagent | P37, *OpiE2* forward | this study | Primer | CAAGCACCTTTATACTCGGTGGCCTC |

(*Continued*)

**Table 1.** (Continued)

| Reagent type (species) or resource | Designation | Source or reference | Identifiers | Additional information |
|---|---|---|---|---|
| Recombinant DNA reagent | P38, *PuroR* reverse 1 | this study | Primer | TTCCAGGAAGGCGGGCACCCC |
| Recombinant DNA reagent | P39, *X1* reverse | this study | Primer | ACTGCAACCCATTCACACTG |
| Recombinant DNA reagent | P41, *SV40* reverse | this study | Primer | CCTCTACAAATGTGGTATGGCTGA |
| Recombinant DNA reagent | P43, *X1* forward 2 | this study | Primer | CCTTGAAGATTGGAATCCATCCAT |
| Recombinant DNA reagent | P47, *Saglin 5'* forward genotyping | this study | Primer | GCTTATAAGAGCGCTACGGGTGTAC |
| Recombinant DNA reagent | P48, *Saglin 3'* reverse genotyping | this study | Primer | CTGTCCCTTCAGCTTCACCAGCTC |
| Recombinant DNA reagent | P49, *Yellow 5'* forward genotyping | this study | Primer | GAACGACAGGTGAGCTGACAGTAAC |
| Recombinant DNA reagent | P50, *Yellow 3'* reverse genotyping | this study | Primer | GATTTGGCGAGATGGTTTGTGCGAC |
| Recombinant DNA reagent | P51, *PuroR* reverse 2 | this study | Primer | GAGCTCGAGTTCAGGCACCGGGCTTGC |
| Recombinant DNA reagent | P52, *EGFP* reverse | this study | Primer | CTTCGGGCATGGCGGACTTG |
| Recombinant DNA reagent | P53, *SV40* forward | this study | Primer | CAGCCATACCACATTTGTAGAG |
| Commercial assay or kit | CloneJET PCR Cloning Kit | ThermoFisher Scientific | Cat#K1231 | / |
| Software, algorithm | Prism 5.0 | GraphPad, San Diego, US | / | https://www.graphpad.com/scientific-software/prism/ |
| Software, algorithm | FIJI | [39] | / | https://imagej.net/software/fiji/ |
| Software, algorithm | MetaMorph | Molecular Devices, San Jose, US | / | https://www.moleculardevices.com/contact |
| Software, algorithm | VectorBase (version 36–55) | [40] | / | https://vectorbase.org/vectorbase/app |
| Software, algorithm | NIS-Elements, F 4.20.01, 64 Bit | Nikon Instruments Inc., Melville, US | / | https://www.microscope.healthcare.nikon.com/products/software/nis-elements |
| Software, algorithm | DNAStar Lasergene, SeqBuilder Pro | DNASTAR, Madison, US | / | https://www.dnastar.com/ |
| Other | Hoechst 33342 | ThermoFisher Scientific | Cat#H3570 | / |

*pJET-3'saglin* and *pJET-3'yellow*. After sequence validation, all building blocks were fused in the correct order in the destination plasmid *pENTR* using GoldenGate assembly together with a building block containing either the fluorescence cassette *pKSB-Lox-3xP3-Lox-EGFP* or *pKSB-Lox-3xP3-EGFP-Lox*, generating *pENTR-Saglin;Lox-3xP3-Lox-EGFP* and *pENTR-Yellow;Lox-3xP3-EGFP-Lox*. Primers are described in **Table 1**.

## Transgenesis of *A. coluzzii*

All plasmids used for DNA injections were purified using Endofree kits (Qiagen). Freshly laid eggs of the docking line X1 or the *vasa-Cas9* line were injected as described [19,45,46]. For

ΦC31-integrase mediated transgenesis, a mix of three plasmids (*pDSAP-aapp-DsRed*, *pDSAP-aapp-hGrx1-roGFP2* and *pDSAP-trio-DsRed*) (~160 ng/μL) and the integrase helper plasmid carrying the *Vasa2-ΦC31-Integrase* gene (Addgene # 62299, ~80 ng/μL) diluted in 0.5x phosphate buffered saline (0.1 mM NaHPO4 buffer, 5 mM KCl, pH 6.8) was injected into approximately 500 embryos. The plasmid *pDSAP-saglin-DsRed* was injected in the same way but into a different batch of eggs. For site-directed knock-in mediated by Cas9, the plasmid solution (*pENTR-Saglin;Lox-3xP3-Lox-EGFP* or *pENTR-Yellow;Lox-3xP3-EGFP-Lox*) (~200–300 ng/μL) was diluted in PBS supplemented with 1 μM Scr7 (APExBio, Houston, Texas) to inhibit non-homologous end joining. Each construct was injected in a separate batch of approximately 500 eggs obtained from *vasa-Cas9* females.

## Salivary gland reporter lines *sag-DsRed*, *trio-DsRed*, *aapp-DsRed* and *aapp-hGrx1-roGFP2*

Following egg micro-injection, surviving pupae were sorted according to sex, hatched in separate cages (G0 generation) and crossed with ~100 mosquitoes of the other sex from a mosquito line expressing *CFP* from the *X1* docking site that was used as a balancer of the new, non-fluorescent but puromycin resistant transgenic insertions at the same locus. Neonate larvae from this cross were treated with puromycin (0.05 mg/mL). G1 pupae were subsequently sex sorted and crossed with CFP positive mosquitoes to expand the population. Adult puromycin resistant G2 females were sorted according to the observed fluorescence phenotype (DsRed or roGFP2) in their salivary glands, reflecting the expression of the *trio-DsRed*, *aapp-DsRed* or *aapp-hGrx1-roGFP2* transgenes and crossed to CFP balancer males. For *sag-DsRed* females, sorting was not necessary, as transgenesis was performed separately by injection of the *pDSAP-saglin-DsRed* plasmid alone. Puromycin-resistant *G3* larvae were raised, intercrossed and the G4 offspring was sorted by COPAS (Marois et al., 2012) for the absence of CFP to select lines homozygous for *sag-DsRed*, *trio-DsRed*, *aapp-DsRed* and *aapp-hGrx1-roGFP2*. Lines were then raised without puromycin treatment. Single females of all generated colonies were genotyped by PCR to monitor correct integration of the transgene into the docking site (**S1A and S1B Fig**). Successfully genotyped mosquito colonies were expanded and used to perform experiments.

## *yellow(-)KI* and *sag(-)KI* lines

Pupae obtained from plasmid injected eggs were sorted according to sex and crossed with 100 *Ngousso* wild type partners. G1 neonate larvae were screened for the expression of *EGFP* using a Nikon SMZ18 fluorescence Stereomicroscope. Selected transgenic larvae were sex sorted at the pupal stage and crossed with an excess of wild type partners. The G2 progenies (from a single positive G1 female for *sag(-)KI* and from 10 positive individuals for *yellow(-)KI*) were COPAS sorted to enrich for the integrated transgene and to remove the *vasa-Cas9* transgene (marked with *DsRed*). The generated *sag(-)KI* colony was kept in equilibrium with the wild type allele, while the *yellow(-)KI* colony was completely homozygotized for the transgene by selecting for mosquitoes with a yellow body color. Homozygous *sag(-)KI* mosquitoes were crossed with the *C2S* line expressing *Cre* in the germline [19] for Cre recombinase-mediated excision of the lox cassette to generate *sag(-)EX* mosquitoes.

## Genotyping of mosquito lines

Genotyping was performed on genomic DNA extracted from single mosquitoes of *aapp-DsRed*, *aapp-hGrx1-roGFP2*, *trio-DsRed*, *sag(-)KI*, *sag(-)EX* and *yellow(-)KI* colonies. Genotyping PCRs were performed using GoTaq Green Mastermix (Promega) (**S1B Fig**), Phusion

polymerase (ThermoFisher Scientific) or the Phire Tissue Direct PCR Master Mix (Thermo-Fisher Scientific) (**S2** and **S7 Figs**) according to manufacturer recommendations. Primers used for genotyping are described in **Table 1**.

## Mosquito infection and evaluation of parasite numbers

Parasites were maintained as frozen stocks at -80˚C or by passage between mice. For this, blood was taken by heart puncture from a donor mouse with a parasitaemia of 3–5%, diluted to $2.10^7$ parasitized erythrocytes in 100 μL 0.9% NaCl that were injected intravenously into a naïve mouse. Parasitemia was monitored via flow cytometry (Accuri C6, BD biosciences) three days later. Mosquitoes were fed for 10–15 min on anesthetized mice with 2.5–3.5% parasite-mia. Blood fed females were sorted and kept on 10% sucrose at 21˚C and 50–70% humidity. Midguts and salivary glands were dissected in PBS under a SZM-2 Zoom Trinocular stereomicroscope (Optika). Images were acquired with the Nikon SMZ18 Stereomicroscope (salivary glands were imaged without and midguts with coverslip) or with a Zeiss LSM 780 confocal microscope (sealed coverslip). To count oocysts, midguts were dissected 7–8 days post infection. Fluorescence images of infected midguts were processed using the watershed segmentation plugin [47] and oocysts were subsequently counted using the „analyze particles"function implemented in Fiji [39]. To count salivary gland sporozoites, salivary glands were collected in PBS 17–18 days post infection and ground with a plastic pestle for one minute. The number of sporozoites was measured using a Neubauer hemocytometer under a light microscope (Zeiss).

## Fluorescence imaging, fluorescence measurements and image analysis

Samples imaged using the Nikon SMZ18 Stereomicroscope were usually acquired using the respective fluorescence filter to visualize expression of *DsRed* or *roGFP2/EGFP* combined with brightfield to visualize the whole specimen. Images were taken using a 63-fold magnification objective (NA 1.4) on a Zeiss Axio Observer Z1 confocal LSM 780 microscope equipped with an Hamamatsu Orca Flash 4.0 V1 and a Yokogawa CSU X-1 camera (e.g. **Figs 4** and **8**), a Zeiss Axiovert 200 fluorescence microscope (only **S2A Fig**) or using an iPhone 6 (only **S9D Fig**). All other images as well as images used for fluorescence measurements were acquired with a Nikon SMZ18 Stereomicroscope. For fluorescence quantifications, images were acquired in the DsRed channel and settings were kept the same for all experiments. For each salivary gland, only a single lobe was measured by analyzing a square of 51×51 pixels covering the central part of the fluorescent area to determine the mean fluorescence intensity. The image processing for all experiments was performed with Fiji [39]. To count cells in median lobes, salivary glands were dissected in PBS supplemented with Hoechst 33342 and incubated for approximately 10 min at room temperature before imaging using a Zeiss LSM 780 confocal microscope. Median lobes free of external tissues and with entire nuclear staining were imaged for GFP and Hoechst 33342 fluorescence. To ensure complete imaging of all nuclei, imaging was started at the bottom of the median lobe and continued in z-direction to the top. The step size was set manually in this process in order not to miss weaker stained nuclei. To determine cell numbers, all images acquired for a given median lobe were opened simultaneously in Fiji and stained nuclei were counted using the "Multipoint Tool". Cells were visually divided into low and high production cells by identifying and defining a transcriptional boundary between the two populations (see **Fig 3C, 3D and 3E**).

## roGFP2 measurements

Dissected salivary glands were treated as in [48]. In brief, tissues were alkylated in 20 mM N-ethyl-maleimide (NEM) for 15 min to preserve the redox state of roGFP2 and then fixed in

paraformaldehyde (4% PFA) for 20 min. In parallel, full oxidation and reduction controls were prepared by incubating some samples in 2 mM diamide or 20 mM dithiothreitol (DTT), respectively, prior to alkylation with NEM and fixation as described before. All compounds were prepared in phosphate-buffered saline. Salivary glands were mounted with VectaShield (Vector Laboratories) and samples were stored at 4˚C in the dark until analysis. Samples were imaged with a Zeiss Axio Observer Z1 confocal LSM780 microscope with GaAsP detector by using a Plan-Apochromat 63x/1.4 oil objective. The fluorescent probe hGrx1-roGFP2 was excited sequentially at 405 and 488 nm, and its emission signal was recorded at 500–530 nm. Images were processed in ImageJ following the protocol described in [49]. In brief, background was subtracted by using the rolling ball procedure set to 50 pixels and images were converted to 8-bit. Then, images were smoothed and stacked. The intensity of the 488 nm image was thresholded with the default settings in black and white; values below threshold were set to "not a number". The ratio image was generated by dividing the 405 nm picture by the 488 nm picture using the "Ratio Plus" plugin, and colored using the lookup table "fire". Subsequently, intensities were normalized to the DTT control.

### *In vivo* imaging

For *in vivo* imaging experiments, infected *yellow(-)-aapp-DsRed* mosquitoes were pre-screened between days 17–21 post infection for the presence sporozoites (GFP fluorescence) at the hinge of the wings and in salivary glands, as well as for cuticular positioning of the salivary glands (DsRed signal) (**S9A and S9B Fig**). After removing their legs, mosquitoes were glued on a microscopy slide using small amounts of UV-light reactive glue (Bondic Pocket) and covered with a coverslip prepared with plasticine feet at each corner to prevent a squeezing of the sample. Corners of the cover slip were coated with nail polish to prevent sliding. Imaging was performed on a Zeiss Axio Observer Z1 confocal LSM780 microscope with GaAsP detector using a Plan-Apochromat 63x/1.4 oil objective. Movies of sporozoites and salivary glands were acquired for a duration of 2–5 minutes and 2–3 seconds between frames using the Metamorph acquisition software.

### Statistical analysis

Statistical analysis was performed using GraphPad Prism 5.0 (GraphPad, San Diego, CA, USA). Data sets were either tested with a one-way ANOVA or a Mann Whitney test. A value of $p < 0.05$ was considered significant.

### Supporting information

**S1 Fig. Genotyping of the salivary gland reporter lines *aapp-DsRed*, *aapp-hGrx1-roGFP2*, *trio-DsRed* and *sag-DsRed*. A)** Schematic representation of salivary gland reporter transgenesis cassettes inserted on chromosome *2L*. Fusion of *attB* and *attP* sites generate *attL* and *attR* sites after integration. The *Lox* site upstream of the transgene is a remnant of the fluorescence cassette initially required to select the docking line *X1*. Note that the illustration is not drawn to scale. **B)** Genotyping of generated transgenic mosquito colonies in comparison to wild type (WT: *G3*) or the parental line *X1*. Five different analytical PCRs were performed. The *5'INT* and *3'INT* PCRs amplify the integration borders upstream and downstream of the transgene, respectively. The presence of the puromycin resistance cassette was tested (*OpiE2-Puro* PCR). To ensure integration of the relevant fluorescence cassette, the PCRs *aapp-DsRed*, *aapp-hGrx1-roGFP2*, *trio-DsRed* and *sag-DsRed* were performed. In addition, a PCR control amplifying the 5'UTR of the *yellow* gene on chromosome *X* was included. Genotyping primers and amplified sequences are indicated with arrowheads and thin black lines, respectively, in

scheme (A). Expected amplicon sizes are indicated under the gel images, (-) indicates that no amplification is expected. Red asterisks mark unspecific amplicons.
(TIF)

**S2 Fig. The intergenic sequence between *AGAP000609* and *AGAP000610* (*saglin*) displays no promoter activity. A)** Images of the apex of the salivary gland median lobe from a *sag-DsRed* and a wild type female (*Ngousso*). Dissected glands were stained with Hoechst 33342 to detect the nuclei of acinar cells. The dotted black line in DsRed images indicates the outline of the salivary gland lobe observed in DIC. Scale bar: 20 μm. **B)** Genomic context of *saglin* (AGAP000610) with neighboring genes. The length of intergenic sequences is indicated above the scheme and the sequence tested for promoter activity located between *AGAP000609* and *saglin* is shown. The stop codon *TAA* of *AGAP000609* and the start codon of *saglin* are highlighted in black. The length of the cloned sequenced intergenic sequence was 220 bp (without stop codon of *AGAP000609* and start codon of *saglin*). Note that the illustration is not drawn to scale.
(TIF)

**S3 Fig. Genotyping of *sag(-)KI*, *sag(-)EX* and *yellow(-)KI* mosquitoes.** *Sag(-)KI* (**A**) and *yellow(-)KI* (**B**) mosquitoes were generated by injecting embryos expressing *Cas9* (*vasa-Cas9*) with plasmids carrying repair templates that contain a *Lox3xP3Lox-EGFP* or a *Lox3xP3-EGFP-Lox* cassette flanked with upstream and downstream sequences of the respective targeted loci, in combination with three guide RNAs specific for the *saglin* (*AGAP000610*) or *yellow* gene (*AGAP000879*). Genotyping of the 5' and 3' integration borders revealed successful integration of the transgenes into *saglin* (**C**) and *yellow* (**D**). The red asterisk marks an unspecific PCR product observed in the wild type control (WT: *Ngousso*). **E)** To investigate the native expression of the *saglin* promoter, the *3xP3* promoter initially used to select transgenic *sag(-) KI* mosquito larvae was removed by Cre-mediated excision. The loss of the *3xP3Lox* sequence in *sag(-)EX* mosquitoes was confirmed by PCR. Expected amplicon sizes are indicated under the gel images, (-) indicates that no amplification is expected (C, D, E).
(TIF)

**S4 Fig. Expression driven by *trio* and *aapp* promoters is female- and adult-specific and the fluorescent reporter accumulates after the emergence of adults. A)** Fluorescence and brightfield images of female / male pairs of *trio-DsRed*, *aapp-DsRed* and *aapp-hGrx1-roGFP2* mosquitoes. Note that all depicted mosquitoes were bred in synchrony and were seven day old (+/- 1 day). Scale bar: 0.5 mm. **B)** Fluorescence and brightfield images of *aapp-DsRed*, *trio-DsRed* and *aapp-hGrx1-roGFP2* pupae in comparison to wild type (*G3*). Dotted circles indicate the putative position of the salivary glands. *Trio-DsRed* pupae with low and high DsRed body fluorescence are shown for comparison. The red arrows indicate DsRed positive ocelli observed in most *trio-DsRed* pupae. Note that the fluorescence and brightfield images do not fully overlap as images pupae were able to move between the two pictures. Scale bar: 0.5 mm. **C)** Brightfield images of *aapp-DsRed* and *trio-DsRed* female salivary glands from day 1 to day 5 after hatching (a single image per day and per line). All images were acquired using the same settings. Scale bar: 250 μm.
(TIF)

**S5 Fig. EGFP and hGrx1-roGFP2 concentrate around nuclei of acinar cells.** Dissected salivary glands from *aapp-hGrx1-roGFP2* and *sag(-)KI* were stained with Hoechst 33342 (DNA) and imaged by confocal microscopy. The *sag(-)KI* line was chosen instead of the *sag(-)EX* line as it displays the same EGFP pattern but with stronger signal. Distal-lateral (top) and median (bottom) lobes are shown for *aapp-hGrx1-roGFP2* and *sag(-)KI*, respectively. Left to right:

DNA and EGFP signals in black on white, combination of both signals in colors, and zooms of cells framed in the previous images. Scale bars: 20 μm except for zooms (10 μm).
(TIF)

**S6 Fig. *Trio-DsRed* pupae display high variability in *DsRed* expression. A)** Representative images of *DsRed* expression patterns in *trio-DsRed* pupae. A subpopulation of pupae displayed high body DsRed fluorescence (images in red frames) while remaining pupae showed no DsRed body fluorescence (black frames). Some pupae displayed fluorescent ocelli and/or fluorescent antennae. The presence of DsRed expression in body, ocelli and antennae occurred in different combinations that are represented by rectangles with different shades and frames. Scale bar: 0.5 mm **B)** Fluorescence patterns in pupae in relation to their sex. Number of pupae indicated above columns. Data pooled from four generations (≥20 pupae per generation). **C)** Proportion of each fluorescence pattern in relation to the whole population. Shading and framing of the different patterns as in (A). **D)** Proportion of pupae with high and low body fluorescence in the *trio-DsRed* colony (F0) and after intercrossing individuals displaying high or low pupal body fluorescence (F1). The number of analyzed individuals is given above each column. **E)** Females hatched from pupae with high body fluorescence analyzed in (D, high fluo F1) were kept separately and crossed to wild type males (*Ngousso*). The progeny of single females was evaluated at the pupal stage for body fluorescence (same color code as D, dashed line indicates 50% of the population).
(TIF)

**S7 Fig. Genotyping of *trio-DsRed low fluo* and *trio-DsRed high fluo* mosquitoes.** Genotyping of a female mosquito homozygous for the *trio-DsRed* low fluorescence transgene (*trio-DsRed low fluo*) taken from the colony generated by the intercross of pupae displaying no body fluorescence (see **Fig 4 –Supplement 1D**) and of a female mosquito heterozygous for the *trio-DsRed* high fluorescence transgene (*trio-DsRed high fluo*) and the wild type allele (*WT*) obtained from family 1 (**Fig 4 –Supplement 1E**). An illustration of the modified *X1* locus containing the *trio-DsRed* transgene in comparison to the unmodified wild type allele is shown on top. Genotyping primers and expected products are shown as black arrows and lines, respectively. The length of the expected PCR products is indicated below the gel images. Note that for the heterozygous female carrying the *trio-DsRed* high fluorescence transgene, the smaller PCR fragment amplifying the unmodified locus dominates while the long PCR fragment representing the transgene containing allele is visible as a faint band at the upper edge (marked by a red arrow). The OpiE2-Puro PCR to verify the presence of the transgene gave several unspecific bands. PCRs were performed on genomic DNA obtained from single mosquitoes.
(TIF)

**S8 Fig. Evaluation of *P. berghei* parasite infection in salivary gland reporter lines and background fluorescence in *yellow(-)KI* mosquitoes.** Oocyst (**A**) and salivary gland sporozoite (spz) (**B**) counts in infected *aapp-DsRed*, *aapp-hGrx1-roGFP2* and *trio-DsRed* females in comparison to *wild type* (*G3*) females. Data pooled from 3–4 experiments generated with three different mosquito generations. The total number of dissected mosquitoes is given above each genotype. Data points represent parasites counted per midgut (A) and mean number of sporozoites per mosquito in independent experiments (B). **C)** Number of salivary gland sporozoites (spz) per oocyst for all three reporter lines in comparison to *wild type* (WT: *G3*). The invasion rate was calculated using the data shown in (A) and (B). Each dot represents the mean of an independent experiment, the median is indicated by a bar. All data were tested for significance using a Kruskal-Wallis test. ns: not significant (p>0.05). **D)** GFP background fluorescence of *yellow(-)KI* in comparison to a wild-type (*Ngousso*) female. Combinatorial transcriptional

activity guided by the *3xP3* and the endogenous *yellow* promoter drive *EGFP* expression in the eye and the ovaries of *yellow(-)KI* females (indicated by red asterisks). No EGFP-like fluorescence was observed in the throat region where the salivary glands localize (indicated by dashed red circle). Scale bar: 1 mm. The image on the right shows EGFP fluorescence from a dissected ovary of a *yellow(-)KI* female at higher magnification.
(TIF)

**S9 Fig. Selection and sample preparation of mosquitoes for *in vivo* imaging and comparison of sporozoite and salivary gland fluorescence signals. A)** Images of two infected mosquitoes with sporozoites (Spz, GFP channel) inside or close to *DsRed*-expressing salivary glands (SG, red channel). The bottom image was acquired with a filter set visualizing signals of DsRed and GFP. Salivary glands in both mosquitoes localised close to the cuticle. Their position is even visible in brightfield as a light red dot. **B)** The position of salivary glands varies between mosquitoes, either deep (rows 1 & 2) or close to the cuticle (rows 3 & 4). Brightfield (left) and DsRed (right) images of the same field of view. Images were acquired with the same settings. Scale bar: 250 μm. **C)** Quantification of salivary gland positioning according to the fluorescence pattern shown in (B). Two mosquito batches were evaluated per time point post emergence (p.e.). The number of evaluated mosquitoes is indicated above each column. **D)** Sample preparation before imaging. Legless mosquitoes were glued on a microscopy slide and covered with a cover slip underlaid with orange plasticine to avoid squeezing. Corner areas were subsequently coated with teal-colored nail polish to prevent the cover slip from moving. **E)** DsRed, GFP and merging the two signals of the image shown in Fig 8A. Note that a significant portion of the DsRed signal is also visible in the GFP channel, whereas the sporozoites are exclusively visible in the GFP channel, potentially indicating spillover of the red signal due to the high DsRed concentration.
(TIF)

**S1 Movie. Sporozoites display active motility inside the mosquito.** Movie showing three sporozoites imaged in close proximity to the salivary gland. Sporozoites appear to be attached to a larger vessel and observed movements might be a combination of active motility and passive hemolymph flow. Imaging performed with a Hamamatsu Orca Flash 4.0 V1 camera using a 63x (NA 1.4) objective. GFP signal of a 3 min movie with 3 seconds between frames.
(AVI)

**S2 Movie. Sporozoites form accumulations close to the salivary gland.** Movie showing several sporozoites directly below the cuticle. Three sporozoites display different patterns of active movement. Imaging performed with a Hamamatsu Orca Flash 4.0 V1 camera using a 63x (NA 1.4) objective. GFP signal of a 2 min and 30 s movie with 2 seconds between frames.
(AVI)

**S1 Appendix. Promoter sequences of the trio, saglin and aapp genes used to generate the mosquito lines aapp-DsRed, aapp-hGrx1-roGFP2, trio-DsRed and sag-DsRed.** Nucleotides highlighted in red are part of the primer sequence used to amplify the promoter sequence. Sequences with a grey background indicate the 5'UTRs of the trio and aapp transcript, respectively.
(DOCX)

## Acknowledgments

We thank Amandine Gautier for help with rearing mosquitoes and technical assistance during infections with *P. berghei*, Sarra Manai for help with experiments, Jean-Daniel Fauny for

assistance during microscopy and the whole mosquito immune responses (MIR) team for discussions and assistance with mosquito breeding. We also thank the CNRS, Inserm and the University of Strasbourg for providing the infrastructure, salaries and for their support.

## Author Contributions

**Conceptualization:** Dennis Klug, Eric Marois, Stéphanie A. Blandin.

**Data curation:** Dennis Klug, Katharina Arnold, Raquel Mela-Lopez.

**Formal analysis:** Dennis Klug, Katharina Arnold, Raquel Mela-Lopez.

**Funding acquisition:** Dennis Klug, Eric Marois, Stéphanie A. Blandin.

**Investigation:** Dennis Klug, Katharina Arnold, Raquel Mela-Lopez, Eric Marois.

**Methodology:** Dennis Klug, Raquel Mela-Lopez.

**Project administration:** Dennis Klug, Eric Marois, Stéphanie A. Blandin.

**Supervision:** Dennis Klug, Eric Marois, Stéphanie A. Blandin.

**Visualization:** Dennis Klug.

**Writing – original draft:** Dennis Klug.

**Writing – review & editing:** Dennis Klug, Katharina Arnold, Raquel Mela-Lopez, Eric Marois, Stéphanie A. Blandin.

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
