## [Decision Letter · Decision Letter 0]

18 Aug 2022

Dear Mr. Klug,

Thank you very much for submitting your manuscript "A toolbox of engineered mosquito lines to study salivary gland biology and malaria transmission" for consideration at PLOS Pathogens. As with all papers reviewed by the journal, your manuscript was reviewed by members of the editorial board and by several independent reviewers. The reviewers appreciated the attention to an important topic. Based on the reviews, we are likely to accept this manuscript for publication, providing that you modify the manuscript according to the review recommendations.

The reviewers of your revised manuscript only recommended minor changes to the text and no additional experiments. Please take these suggestions into consideration and adjust the manuscript accordingly. 

Sincerely,

Kirk W. Deitsch

Section Editor

PLOS Pathogens

Kirk Deitsch

Section Editor

PLOS Pathogens

Kasturi Haldar

Editor-in-Chief

PLOS Pathogens

orcid.org/0000-0001-5065-158X

Michael Malim

Editor-in-Chief

PLOS Pathogens

orcid.org/0000-0002-7699-2064

Reviewer Comments (if any, and for reference):

Reviewer's Responses to Questions

**Part I - Summary**

Reviewer #1: Klug et al. provides a detailed account of the generation of fluorescent reporter lines in the African malaria mosquito Anopheles coluzzii using the salivary gland-specific promoters of the anopheline antiplatelet protein (AAPP), the triple functional domain protein (TRIO) and saglin (SAG) genes. Promoter activity was specifically observed in the distal-lateral lobes or in the median lobe of the salivary gland. Besides a comparison of the expression patterns of the selected promoters, the fluorescent probes allowed evaluation of the inducibility of these three promoters upon blood feeding and to measure intracellular redox changes. They also combined the aapp-DsRed fluorescent reporter line with a pigmentation-deficient yellow(-) mosquito mutant in order to assess the feasibility of in vivo microscopy of parasitized salivary glands. This combination allowed location of the salivary gland through the cuticle and imaging of individual sporozoites in vivo, which facilitates live imaging studies of salivary gland colonization by Plasmodium sporozoites.

Strengths: The manuscript is well written, and the conclusions presented are both appropriate in scope and well-justified by the data presented and described within. The edits to the title greatly improve the reader's expectations for the scope of this research, and the tools that have been generated by these authors are both very useful and serve as an indicator of the current state of this field (goals and future directions).

Weaknesses: The limitations of this study result primarily from the relative deficiencies of standard microscope approaches and the mosquito as a research organism. Great future efforts will be needed to fully utilize molecular genetics approaches to better understand the journey of the sporozoite through into, through, and out of, the mosquito salivary glands. Specifically regarding this study, not using human malaria parasites is a key weakness, but understandable at this time. The authors provided an appropriate justification for their choices of mosquito and parasite species.

Novelty/Significance: The authors applied the most appropriate, cutting edge technologies (e.g. CRISPR/Cas9 editing) in meaningful ways in order to address the aim of this study. They also utilized meaningful historical approaches (e.g. yellow mutants) in innovative ways. While not exclusively novel, per se, these authors have conducted a well-designed and supported research study of great significance to the vector biology community that provides further insights into mosquito SG-Plasmodium interactions and provides the impetus for further methodological improvements and deeper future analyses.

Execution: The data presented by these authors is complete and rigorous, meeting or exceeding expectations for this level of inquiry. The figures are informative and presented clearly, labeled well, and represent sufficient experimental replicates to validate their results.

Scholarship: The authors demonstrate strong command of the historical and most current scientific literature relevant to their study. Appropriate citations have been included and are representative of the studies most closely linked to the introduction and conclusions of this work. The authors did a masterful job of reworking the introduction to better frame the scope of this research.

Reviewer #2: This is a revised version of a manuscript that I have previously reviewed for Review Commons (as reviewer #1). This manuscript reports the generation and characterization of transgenic lines in the African malaria mosquito Anopheles coluzzii, expressing fluorescent proteins in the salivary glands, and their potential use for in vivo imaging of Plasmodium sporozoites.

As already indicated in my previous report, this is a nice piece of work describing a clear technical advance. The generation of the transgenic mosquito lines is elegant and state-of-the art, and the new reporter lines are thoroughly characterized. In their revised version, the authors addressed all my comments. In particular, they provide new imaging data supporting the proof-of-concept that sporozoites can be visualized in live mosquitoes. Future studies using these new transgenic mosquito lines may bring novel biological insights into salivary gland colonization by malaria parasites.

Reviewer #3: The manuscript describes generation of fluorescent reporter lines in A. coluzzii using salivary gland-specific promoters such as AAPP, TRIO and SAG. The characterization of these lines from larvae to mosquitoes are well described and in-depth description of the fluorescent reporter expression in salivary glands are informative. Fluorescence signal of aapp-DsRed line localized in the distal lateral lobe, and trio-DsRed, sag (-)KI and sag(-)EX signal localized in the median lobe. More specifically, in the median lobe, stronger signal of sag (-)KI was observed in the base and trio-DsRed signal was stronger in the apex. While trio-DsRed fluorescence intensity was not blood feeding dependent, aapp-DsRed and sag(-)EX fluorescence intensity increased upon blood feeding. In addition, the authors showed the evidence of a functional hGrx1-roGFP2 probe and it is interesting that P. berghei infection does not affect redox status of salivary glands. Finally, authors performed in vivo imaging and successfully visualized the salivary gland and sporozoites. In vivo imaging requires further development as authors discussed, and together with current development of imaging techniques (e.g. https://doi.org/10.1101/2021.10.15.463778 in bioRxiv), these lines will contribute to study the sporozoite-salivary gland interaction in the future.

**Part II – Major Issues: Key Experiments Required for Acceptance**

Reviewer #1: I did not identify any major issues with this revised manuscript.

Reviewer #2: I find it surprising that despite heavy sporozoite loads in the glands only a few sporozoites could be imaged (Fig 8 and Fig 7supplt2). Is it merely a technical issue due to the difficulties in imaging deep tissues (such as salivary glands) in live mosquitoes? The authors should discuss further how imaging could be improved, in terms of microscope type, objectives, etc... This could be instrumental for full exploitation of the new tools they have generated in this work.

Reviewer #3: No new experiments required.

**Part III – Minor Issues: Editorial and Data Presentation Modifications**

Reviewer #1: The figures are largely very clear. I identified two areas for minor improvements.

In Fig. 7C - creating a second version of lower right panel with dashed outline around the presumptive SGs might make this figure easier to interpret than arrows for non-experts.

In Fig. 8A quantification, comma/period use is inconsistent within the graph (right side).

Reviewer #2: -I feel it would be more logical to place lines 380-390 after lines 391-397.

-Line 384: typo “surprisingly”

-In fig7B there seems to be an auto-fluorescent structure in the GFP channel, which does not overlap with the salivary gland DsRed signal. This is in contradiction with Fig7-supplt1D where no such fluorescence is observed in the throat region. Are these really Pb sporozoites? Perhaps the authors could select a more convincing image.

-Line 409-410: invasion of the salivary glands begins before day 17

Reviewer #3: Line 101

Previous description on sporozoites in the cavities of the acinar cells used “bundles” (PMID: 7866385 also in Wells and Andrew, 2019). I would suggest to change “stacks” to “bundles” and add the reference (PMID: 7866385).

Line 271-278

The description of number of cells in median lobe and quantification of high-low signal of is informative. However, I did not find the detailed method. Please add following information to Materials and Methods section.

• how you acquired z-stacks (increment)

• description of how you quantify cells

• description of how you determine high-low promoter activity

Figure 6C

It is an interesting observation that there is no change in redox state in the salivary glands upon infection with P. berghei. Please add the number of experiments performed in the figure legend.

Line 460-462

Please clarify these line. Especially, “the protein (saglin) has been shown to affect colonization of the salivary glands by Plasmodium sporozoite (Ghosh and Jacobs-Lorena, 2009)”. I assume these lines refer to the TRAP-saglin interaction during sporozoite invasion of salivary glands. If so, please change the reference to (Gosh, A.K. et al, 2009).

PLOS authors have the option to publish the peer review history of their article (what does this mean?). If published, this will include your full peer review and any attached files.

Reviewer #1: No

Reviewer #2: No

Reviewer #3: No

Figure Files:

Data Requirements:

Reproducibility:

References:

---

## [Editor Report · Decision Letter 1]

12 Sep 2022

Dear Mr. Klug,

We are pleased to inform you that your manuscript 'A toolbox of engineered mosquito lines to study salivary gland biology and malaria transmission' has been provisionally accepted for publication in PLOS Pathogens.

Best regards,

Kirk W. Deitsch

Section Editor

PLOS Pathogens

Kirk Deitsch

Section Editor

PLOS Pathogens

Kasturi Haldar

Editor-in-Chief

PLOS Pathogens

orcid.org/0000-0001-5065-158X

Michael Malim

Editor-in-Chief

PLOS Pathogens

orcid.org/0000-0002-7699-2064
---

## [Editor Report · Acceptance letter]

28 Sep 2022

Dear Mr. Klug,

We are delighted to inform you that your manuscript, "A toolbox of engineered mosquito lines to study salivary gland biology and malaria transmission," has been formally accepted for publication in PLOS Pathogens.

Best regards,

Kasturi Haldar

Editor-in-Chief

PLOS Pathogens

orcid.org/0000-0001-5065-158X

Michael Malim

Editor-in-Chief

PLOS Pathogens

orcid.org/0000-0002-7699-2064